# A Bayesian Approach To Analysing Training Data Attribution In Deep Learning

**Elisa Nguyen**
Tübingen AI Center
University of Tübingen

**Minjoon Seo**
KAIST AI

**Seong Joon Oh**
Tübingen AI Center
University of Tübingen

## Abstract

Training data attribution (TDA) techniques find influential training data for the model's prediction on the test data of interest. They approximate the impact of down- or up-weighting a particular training sample. While conceptually useful, they are hardly applicable to deep models in practice, particularly because of their sensitivity to different model initialisation. In this paper, we introduce a Bayesian perspective on the TDA task, where the learned model is treated as a Bayesian posterior and the TDA estimates as random variables. From this novel viewpoint, we observe that the influence of an individual training sample is often overshadowed by the noise stemming from model initialisation and SGD batch composition. Based on this observation, we argue that TDA can only be reliably used for explaining deep model predictions that are consistently influenced by certain training data, independent of other noise factors. Our experiments demonstrate the rarity of such noise-independent training-test data pairs but confirm their existence. We recommend that future researchers and practitioners trust TDA estimates only in such cases. Further, we find a disagreement between ground truth and estimated TDA distributions and encourage future work to study this gap. Code is provided at `https://github.com/ElisaNguyen/bayesian-tda`.

## 1 Introduction

Understanding how machine learning models arrive at decisions is desirable for social, legal and ethical reasons, particularly for opaque deep learning models [1]. One approach to explanations is the data-centric approach of training data attribution (TDA). As the name suggests, TDA finds attributing training samples for a model decision, uncovering which part of the training data is relevant. The attribution $\tau$ of a training sample $z_j$ on another sample $z$ is usually defined as the change of model loss $\mathcal{L}$ on $z$ when the model is retrained without $z_j$ [2]:

$$\tau(z_j, z) := \mathcal{L}(z; \theta_{\backslash j}) - \mathcal{L}(z; \theta) \tag{1}$$

where $\theta$ is a model trained on the entire dataset $\mathcal{D}$ and $\theta_{\backslash j}$ is a model trained on the same set without $z_j$. Since the direct computation of Equation 1 is expensive, various TDA techniques for approximating the quantity have been proposed, such as influence functions [3] or TracIn [4]. Their approximations are often based on some form of inner product between the parameter gradients $\nabla_\theta \mathcal{L}(z; \theta)$ and $\nabla_\theta \mathcal{L}(z_j; \theta)$.

Knowing how training samples attribute to a model decision provides an actionable understanding of the training data distribution, especially in cases of model error. TDA methods can identify the training samples that are most relevant to an error and therefore enable users to understand why the error occurred (e.g. due to domain mismatch of test and training data or wrongly labelled training data) [3]. Additionally, TDA gives them the tool to address the errors by e.g. changing the model directly through the training data. Even in non-erroneous cases, understanding the attributing training

37th Conference on Neural Information Processing Systems (NeurIPS 2023).

data may enable users affected by model decisions to contest the decisions if the attributing training data is noisy or of low quality [5].

At the same time, TDA methods, especially influence functions [3], have been criticised for their fragility when applied to deep models [6, 7, 8]. The main reasons are model complexity and the stochasticity of deep model training. While the former poses a challenge specifically for influence functions as they rely on strong convexity assumptions, the latter is a more general challenge [6, 9]. The randomness inherent to the training process does not only lead to variation in the learned model parameters but also in TDA scores, which makes them untrustworthy. Hence, K & Søgaard [9] recommend using expected TDA scores for increased stability.

We argue that solely considering the expectation is not sufficient to ensure the reliability of TDA but requires inspecting the variance, too. We introduce a Bayesian perspective on the TDA task, noticing that there is no deterministic mapping from a dataset $\mathcal{D}$ to the corresponding model $\theta$ for deep neural networks. The learned model depends on the initialisation and batch composition in the stochastic gradient descent (SGD) optimiser. We capture the resulting randomness via Bayesian model posterior $p(\theta|\mathcal{D})$ over the parameter space [10, 11, 12]. In turn, the TDA estimate (Equation 1) is a random variable that depends on two posteriors, $p(\theta|\mathcal{D})$ and $p(\theta_{\setminus j}|\mathcal{D}_{\setminus j})$.

This viewpoint leads to a few insights into the practical usage and evaluation of TDA techniques. We confirm quantitatively that the ground-truth influence $\tau(z_j, z)$ is often dominated by the noise: $\sqrt{\mathrm{Var}(\tau)} > \mathbb{E}|\tau|$. We argue that it is practically difficult to apply any TDA technique on pairs $(z_j, z)$ whose ground-truth attributions $\tau(z_j, z)$ are noisy in the first place. Likewise, any evaluation of TDA methods on such high-variance pairs would not be reliable.

Nonetheless, we are optimistic that TDA techniques are useful in practice, particularly for train-test pairs with high signal-to-noise ratios: $\sqrt{\mathrm{Var}(\tau)} \ll \mathbb{E}|\tau|$. We observe that such pairs are rare but consistently present in multiple experiments. We recommend that researchers and practitioners confine their usage to scenarios where the signal-to-noise ratios are expected to be large enough.

Our contributions are as follows: (1) Bayesian formulation of the training data attribution (TDA) task. (2) Observation that the ground-truth TDA values are often unreliable and highly variable. (3) Recommendation for the community to use the TDA tools only when the expected noise level is low. (4) Experimental analysis of the contributing factors to the variance of ground-truth TDA values. (5) Observation that the TDA estimation methods capture local changes in the model with regard to the counterfactual question of "retraining without training sample $z_j$", while LOO retraining itself results in a more global change through the training procedure.

## 2 Background

We cover the background materials for the paper, including the concept, method, and evaluation of training data attribution (TDA) methods and Bayesian deep learning.

### 2.1 Training data attribution (TDA)

We introduce the TDA task, a few representative TDA methods, and existing evaluation strategies.

**TDA task.** Given a deep model $f_\theta$ parametrised by $\theta$, a training set $\mathcal{D} := \{z_1, \cdots, z_N\}$, and a test sample $z$, one is interested in the impact of a training sample $z_j$ on the model's behaviour on the test sample $z$. In the TDA context, one is often interested in the counterfactual change in the loss value for $z$ after **leave-one-out (LOO)** training, when $z_j$ is excluded from the training set (Equation 1). TDA has been considered in different use cases, such as understanding the bias in word embeddings [13], fact tracing in language model outputs [14] and measuring the robustness of model predictions [5].

**TDA methods.** The conceptually most straightforward way to compute the difference due to LOO training (Equation 1) is to compute it directly. However, this is computationally expensive, as it involves the learning algorithm for obtaining $\theta_{\setminus j}$ for every $j$. This gives rise to various TDA techniques that find *approximate* estimates $\tau'(z_j, z)$ of LOO. A prominent example of such approximation is the **influence function (IF)** method [3] based on [15]. Under strong smoothness assumptions, they

approximate Equation 1 by:

$$\tau'(z_j, z) := -\nabla_\theta \mathcal{L}(z; \theta)^\top H_\theta^{-1} \nabla_\theta \mathcal{L}(z_j; \theta) \tag{2}$$

where $\nabla_\theta \mathcal{L}(z; \theta)$ and $\nabla_\theta \mathcal{L}(z_j; \theta)$ refer to the parameter gradients of $f_\theta$ for $z$ and $z_j$ respectively. Recognising the difficulty of scaling up the inverse Hessian computation $H_\theta^{-1}$ and the high dimensionality of operations in Equation 2, subsequent papers have proposed further approximations to speed up the computation [16, 17]. Charpiat *et al.* [18] have analysed the influence of $z_j$ on $z$ by dropping the need to compute the Hessian and formulating influence as the loss change when an **additional training step (ATS)** on $z_j$ is taken:

$$\tau(z_j, z) := \mathcal{L}(z; \theta_{+j}) - \mathcal{L}(z; \theta) \tag{3}$$

where $\theta_{+j}$ is a learned model parameter with $\mathcal{D}$ and an additional step on $z_j$. They propose two approximations:

$$\textbf{Grad-Dot (GD)}: \quad \tau'(z_j, z) := \nabla_\theta \mathcal{L}(z_j; \theta)^\top \nabla_\theta \mathcal{L}(z; \theta) \tag{4}$$

$$\textbf{Grad-Cos (GC)}: \quad \tau'(z_j, z) := \frac{\nabla_\theta \mathcal{L}(z_j; \theta)}{\|\nabla_\theta \mathcal{L}(z_j; \theta)\|}^\top \frac{\nabla_\theta \mathcal{L}(z; \theta)}{\|\nabla_\theta \mathcal{L}(z; \theta)\|} \tag{5}$$

This method is closely linked to **TracIn** [4] which computes the Grad-Dot not just at the end of the training, but averages the regular Grad-Dot similarities throughout the model training iterations. We note later in our analysis that within our Bayesian treatment of TDA, the TracIn method coincides conceptually with the Grad-Dot method. In our analysis, we study the sensitivity of LOO and the above TDA methods against noise.

**TDA evaluation.** The primal aim of TDA methods is to measure how well they approximate the ground-truth LOO values. This is often done by measuring the correlation between the estimates from each TDA method and the ground-truth LOO values (Equation 1) [3, 6, 7, 9]. They use either a linear (Pearson) correlation or a rank (Spearman) correlation over *a small number* of train-test sample pairs $(z_j, z)$ due to the computational burden of computing the actual LOO values, especially for larger models. Usually, a few samples $z$ are chosen for a comparison against LOO, e.g. Koh & Liang [3] report results for one $z$ and Guo *et al.* [16] for 10 samples $z$. In some cases, the ground-truth LOO is obtained by computing the change in loss after training further from the learned model parameters, e.g. [3]. Some works have adopted indirect evaluation metrics such as the retrieval performance of mislabelled or poisoned training data based on the TDA estimates [3, 19, 16, 9, 17]. In this work, we adopt the Pearson and Spearman correlation metrics and discuss ways to extend them when the target (LOO from same initialisation) and estimates (TDA) are both random variables.

## 2.2 Bayesian deep learning.

Bayesian machine learning treats the learned model as a posterior distribution over the parameter space, rather than a single point:

$$p(\theta|\mathcal{D}) = p(\mathcal{D}|\theta)p(\theta)/p(\mathcal{D}). \tag{6}$$

Bayesian ML nicely captures the intuition that the mapping from a training set $\mathcal{D}$ to the learned model $p(\theta|\mathcal{D})$ is not a deterministic mapping, especially for non-convex models like deep neural networks (DNNs). Depending on the initialisation, among other factors, DNN training almost always learns vastly different parameters.

The estimation of the true posterior is indeed difficult for complex models like DNNs. The field of Bayesian deep learning is dedicated to the interpretation of certain random elements in DNN training as sources of randomness for the approximated Bayesian posteriors. For example, if Dropout [20] is used for training a model, it may be used at test time to let users sample $\theta$ from the posterior distribution $p(\theta|\mathcal{D})$ [21]. More generally used components like stochastic gradient descent (SGD) have also been interpreted as sources of randomness. The random walk induced by SGD iterations in the parameter space can be viewed as a Markov Chain Monte-Carlo sampler from the posterior distribution, after a slight modification of the optimisation algorithm (Stochastic Gradient Langevin Dynamics [10]). Similarly, the last few iterations of the vanilla SGD iterations may also be treated as samples from the posterior, resulting in more widely applicable Bayesian methods like Stochastic Weight Averaging (SWA) [12, 22]. Finally, the random initialisation of DNNs has also been exploited for modelling posterior randomness; training multiple versions of the same model with different initial parameters may be interpreted as samples from the posterior [11]. We show in the next section how the Bayesian viewpoint will help us model the sources of stochasticity for TDA estimates.

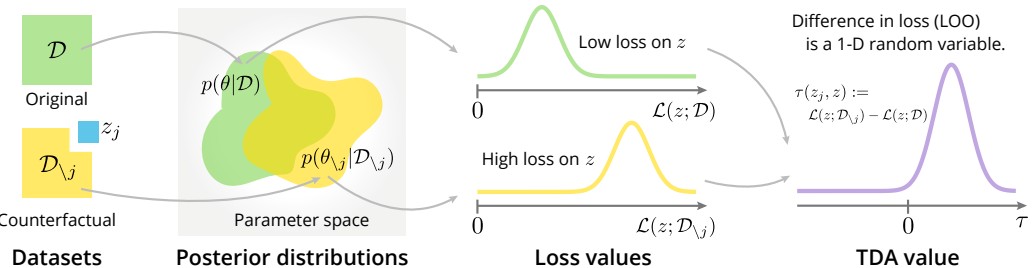

Figure 1: **A Bayesian interpretation** of training data attribution (TDA).

## 3 A Bayesian perspective on training data attribution

Training data attribution (TDA) $\tau(z_j, z)$ is defined as the attribution of one training sample $z_j$ to another sample $z$ in terms of how a target metric like the loss of a sample $\mathcal{L}(z, \theta)$ changes when the model is trained without $z_j$ (Equation 1). We note here that according to the definition, we are interested in the impact of the change in the *dataset* from $\mathcal{D}$ to $\mathcal{D}_{\setminus j}$, rather than the change in the model parameter. From a Bayesian perspective, a change in the training dataset leads to a shift in the posterior distribution, $p(\theta|\mathcal{D}) \to p(\theta_{\setminus j}|\mathcal{D}_{\setminus j})$, leading to the definition of TDA as a random variable:

$$\tau(z_j, z|\mathcal{D}) := \mathcal{L}(z; \mathcal{D}_{\setminus j}) - \mathcal{L}(z; \mathcal{D}) = \mathcal{L}(z; \theta_{\setminus j}|\mathcal{D}_{\setminus j}) - \mathcal{L}(z; \theta|\mathcal{D}) \qquad (7)$$

where $\theta \sim p(\theta|\mathcal{D})$ and $\theta_{\setminus j} \sim p(\theta_{\setminus j}|\mathcal{D}_{\setminus j})$. This interpretation is more natural, given the non-uniqueness of the mapping from a training dataset $\mathcal{D}$ to the optimal model parameter $\theta$ for general, non-convex models like DNNs. Alternatively, one could treat the model built from $\mathcal{D}$ as a fixed variable rather than a posterior as TDA is applied to a specific model in practice. The change of dataset from $\mathcal{D}$ to $\mathcal{D}_{\setminus j}$ however still introduces ambiguity in $\theta_{\setminus j}$, which is captured in the Bayesian posterior $p(\theta_{\setminus j}|\mathcal{D}_{\setminus j})$. In this study, we use the probabilistic formulation of TDA in Equation 7.

**Sampling TDA values.** One could plug in various Bayesian DL techniques (§2.2) to compute samples of $p(\theta|\mathcal{D})$, which can be used to get the samples of $\tau(z_j, z)$. In our work, we use the Stochastic Weight Averaging (SWA) [12, 22] and Deep Ensemble (DE) [11] which are applicable to a wide class of deep models. More specifically, we obtain $T$ samples $\theta^{(1)}, \cdots, \theta^{(T)} \sim p(\theta|\mathcal{D})$ either by taking the last $T$ model checkpoints of the SGD iterations (SWA) or by taking the last model checkpoints from $T$ different model initialisations (DE). The same is done for the counterfactual posterior $\theta_{\setminus j}^{(1)}, \cdots, \theta_{\setminus j}^{(T)} \sim p(\theta_{\setminus j}|\mathcal{D}_{\setminus j})$. This results in a mixture-of-Gaussian posterior, where DE samples correspond to centroids of the distribution. Our sampling approach is thus a version of stratified sampling, where the number of samples $T$ from a centroid is fixed and sampled IID.

**Statistical analysis on TDA.** The simplest statistics for the TDA $\tau(z_j, z)$ are the mean and variance:

$$\mathbb{E}[\tau(z_j, z)] = \frac{1}{T} \sum_t \mathcal{L}(z; \theta_{\setminus j}^{(t)}) - \mathcal{L}(z; \theta^{(t)}) \qquad (8)$$

$$\text{Var}[\tau(z_j, z)] = \frac{1}{T^2} \sum_{t,t'} \left( \mathcal{L}(z; \theta_{\setminus j}^{(t)}) - \mathcal{L}(z; \theta^{(t')}) - \mathbb{E}[\tau(z_j, z)] \right)^2 \qquad (9)$$

Our main interest lies in whether the influence of the training data $z_j$ on the test data $z$ is statistically significant and not dominated by the inherent noise of deep model training. For this purpose, we design a Student t-test [23] for quantifying the statistical significance. Our null and alternative hypotheses are:

$$H_0 : \mu = 0 \qquad H_1 : \mu \neq 0. \qquad (10)$$

We consider the test statistic based on sample mean and variance:

$$t = \frac{\mu - \mathbb{E}[\tau(z_j, z)]}{\sqrt{\text{Vars}[\tau(z_j, z)]/T^2}}. \qquad (11)$$

Vars refers to the sample variance where the denominator in Equation 9 is $T^2 - 1$ instead. We report the significance of the absolute TDA $|\tau(z_j, z)|$ for every train-test pair $(z_j, z)$ by computing the p-value corresponding to the t-test statistic. The greater the p-value is, the greater the dominance of noise is for the TDA estimate.

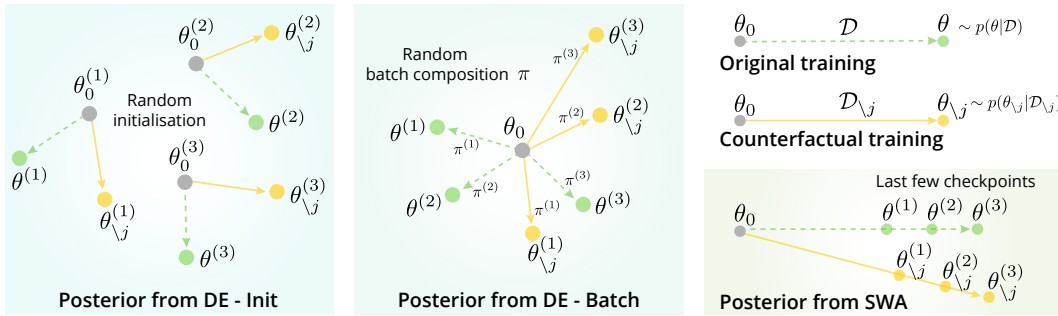

Figure 2: **Sources of randomness for Bayesian posteriors.** In each case, the training starts from initialisation $\theta_0$. Depending on whether $z_j$ is included in the training data, one has either samples from the original posterior $p(\theta_{\backslash j}|\mathcal{D})$ or from the counterfactual posterior $p(\theta_{\backslash j}|\mathcal{D}_{\backslash j})$. For deep ensemble [11], the randomness stems either from random initialisation (DE-Init) or from SGD batch composition (DE-Batch). For stochastic weight averaging (SWA) [12, 22], last few checkpoints of the training are treated as posterior samples.

**TDA methods likewise estimate random quantities.** Approximate TDA methods like influence functions (IF), Grad-Dot, and Grad-Cos (§2.1) also predict random quantities $\tau'(z_j, z)$. For example, IF predicts $\tau(z_j, z) \approx \tau'(z_j, z) := -\nabla_\theta \mathcal{L}(z_j; \theta)^\top H_\theta^{-1} \nabla_\theta \mathcal{L}(z; \theta)$, where one may sample $\theta$ from the posterior $p(\theta|\mathcal{D})$. We note that IF suffers theoretical issues in its application to deep models, as convexity assumptions are not met. In practice, estimation algorithms make use of a damping term to ensure the positive definiteness of the inverse Hessian. Through a Bayesian lens, the damping term could be seen as an isotropic Gaussian prior centred at the origin. Similar statistical analyses on the TDA estimations can be performed as above, including the mean and variance computations and statistical testing for the significance of influence.

**Evaluating TDA as a random variable.** Previously, the LOO-based TDA values $\tau(z_j, z)$ and the estimates from various approximate TDA methods $\tau'(z_j, z)$ are compared via correlation measures like Pearson or Spearman. Our treatment of those quantities as 1-D random variables poses a novel challenge for evaluation because there exists no inborn notion of ordering among 1-D random variables. We address the challenge by examining the approximation ability of TDA methods for both the first and second moments of the true TDA values $\tau(z_j, z)$. More specifically, we compute the Pearson and Spearman correlation for both the mean (Equation 8) and variance (Equation 9) between the ground-truth $\tau(z_j, z)$ and estimated TDA $\tau'(z_j, z)$ values across multiple train-test pairs $(z_j, z)$.

## 4 Experiments

We introduce our experimental settings, present analyses on factors contributing to the reliability of TDA values, compare TDA methods, and draw suggestions on the evaluation practice of TDA.

### 4.1 Implementation details

We illustrate the specific details of our implementation. See the Appendix for further information.

**TDA methods.** We study different TDA methods from a Bayesian perspective. We test the methods introduced in §2.1 for estimating TDA: **influence functions (IF)** [3], **Grad-Dot (GD)** and **Grad-Cos (GC)** [18]. We use the PyTorch implementation of IF from Guo *et al.* [16] and modify it for our models. As the ground-truth target, we consider **Leave-one-out training (LOO)** [3]. For LOO, we remove of $z_j$ from the training set $\mathcal{D}$ by zeroing out the weight for sample $z_j$ towards the loss. Additionally, we include Charpiat *et al.*'s [18] notion of TDA that a training data point $z_j$ attributes more if an **additional training step (ATS)** on it changes the test loss more significantly.

**Inducing randomness in posterior** $p(\theta|\mathcal{D})$**.** In §2.2, we have introduced the interpretation of various elements around model training as sources of randomness for Bayesian posterior. We summarise our methods for inducing randomness in Figure 2. We use the notion of the Deep Ensemble (DE) [11] to sample from the posterior. In a variant of DE with the initialisation as the source of randomness (**DE-Init**), we train each of $T_{\text{DE}}$ randomly initialised parameters $\theta_0^{(t)}$ on either $\mathcal{D}$ or $\mathcal{D}_{\backslash j}$. The resulting parameter sets, $\theta^{(t)}$ and $\theta_{\backslash j}^{(t)}$, are treated as samples from respective

Table 1: **Stability of TDA estimates.** We report p-values for the ground-truth TDA $\tau(z_j, z)$ (LOO) and the estimated TDA values $\tau'(z_j, z)$ (rest 4 columns). The p-values are averaged across all train-test pairs $(z_j, z)$. We use the CNN model throughout.

| Data | Randomness | LOO | ATS | IF | GD | GC |
|---|---|---|---|---|---|---|
| MNIST3 | SWA+DE-Init | 0.331 | 0.254 | 0.352 | 0.363 | 0.003 |
| | SWA+DE-Batch | 0.025 | 0.039 | 0.000 | 0.000 | 0.000 |
| CIFAR10 | SWA+DE-Init | 0.692 | 0.437 | 0.575 | 0.587 | 0.356 |
| | SWA+DE-Batch | 0.487 | 0.296 | 0.484 | 0.517 | 0.236 |

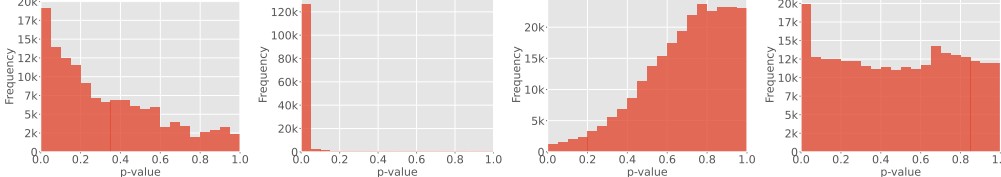

| MNIST3, SWA+DE-I | MNIST3, SWA+DE-B | CIFAR10, SWA+DE-I | CIFAR10, SWA+DE-B |

Figure 3: **Stability of TDA estimates per train-test pair.** Distribution of p-values for ground-truth TDA (LOO) for different experiments.

posteriors. We also consider the batch composition in stochastic gradient descent (SGD) as the source of randomness (**DE-Batch**). In this case, we train from one initial parameter $\theta_0$ with $T_{\text{DE}}$ different random shuffles $\pi^{(t)}$ of the training sets $\mathcal{D}$ and $\mathcal{D}_{\setminus j}$. This results in two sets of samples from the original and counterfactual posteriors. We increase the number of samples by taking the last $T_{\text{SWA}}$ checkpoints as the Stochastic Weight Averaging (**SWA**) samples [12, 22]. For Grad-Dot, this coincides with the definition of TracIn [4] as we average the dot products across checkpoints. In total, we take $T = T_{\text{DE}} \times T_{\text{SWA}} = 10 \times 5$ samples from $p(\theta|\mathcal{D})$ and $p(\theta_{\setminus j}|\mathcal{D}_{\setminus j})$ to estimate $\tau(z_j, z)$.

**Datasets $\mathcal{D}$.** To enable an exhaustive analysis of every train-test pair $(z_j, z)$, we define smaller datasets. We use variants of MNIST [24] limited to three classes (MNIST3), and CIFAR10 [25]. For MNIST3, we sample a training set of size 150 and a test set of size 900, i.e. 135,000 train-test pairs. For CIFAR10, we define the training and test set at size 500, i.e. 250,000 train-test pairs.

**Models.** We consider two types of image classifiers, visual transformers (ViT, [26]) and convolutional neural networks [27], where we primarily study a two-layer (CNN2-L). We also include a three-layer version (CNN3-L) to study the factor of model complexity. For ViT variants, instead of full finetuning, we use LoRA adapter layers [28] to minimise the number of parameters being tuned. The number of trainable parameters of ViT+LoRA (597,514) is comparable to CNN3-L (620,362).

## 4.2 Reliability of TDA evaluation

We assess the reliability of TDA evaluation by measuring the degrees of noise in both the ground-truth TDA (LOO) $\tau(z_j, z)$ and the estimated TDA $\tau'(z_j, z)$. The noise level is measured with the p-value of the Student-t hypothesis testing to determine if the absolute TDA values are significantly greater than the sample noise (§3).

We report the results in Table 1. Generally, we observe many TDA measurements, ground-truth and estimations likewise, are unstable with non-significant p-values ($> 0.05$). In particular, even the ground-truth LOO shows p-values of 0.331 on MNIST3 and 0.692 for CIFAR10 (SWA+DE-Init). In these cases, the noise effectively dominates the signal and any evaluation that does not consider the variance in the posterior $p(\theta|\mathcal{D})$ is likely to be misleading. This confirms the reports in [9] that TDA values are sensitive to model initialisation.

TDA methods often show similar levels of instability. For example, the IF attains p-values 0.352 and 0.575 on MNIST3 and CIFAR10, respectively, roughly matching the LOO case. Grad-Cos is an exception: it attains lower p-values than the other TDA methods (0.003 and 0.356 for MNIST3 and CIFAR10, respectively). We interpret this as an overconfident TDA estimation. Practitioners shall be wary of using TDA methods that are unreasonably stable when the ground-truth TDA itself is not.

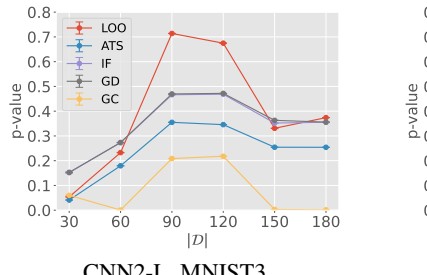 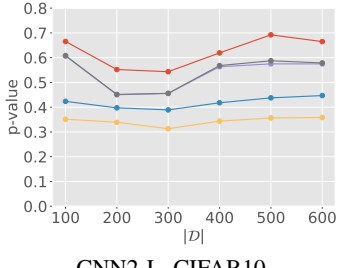

CNN2-L, MNIST3  CNN2-L, CIFAR10

Figure 4: **Impact of training data size.** Mean p-values of TDA methods with randomness induced by SWA+DE-Init.

Table 2: **Impact of model complexity.** Mean p-values of the ground-truth TDA $\tau(z_j, z)$ (LOO) and estimated TDA values $\tau'(z_j, z)$ (the other 4 columns) with randomness induced by SWA+DE-Batch for MNIST3 and DE-Batch for CIFAR10.

| Model | Data | LOO | ATS | IF | GD | GC |
|---|---|---|---|---|---|---|
| CNN2-L | MNIST3 | 0.025 | 0.039 | 0.000 | 0.000 | 0.000 |
| CNN3-L | MNIST3 | 0.370 | 0.368 | 0.464 | 0.470 | 0.005 |
| ViT+LoRA | MNIST3 | 0.786 | 0.573 | 0.369 | 0.365 | 0.093 |
| CNN2-L | CIFAR10 | 0.623 | 0.374 | 0.535 | 0.534 | 0.314 |
| CNN3-L | CIFAR10 | 0.687 | 0.432 | 0.579 | 0.581 | 0.365 |
| ViT+LoRA | CIFAR10 | 0.777 | 0.766 | 0.686 | 0.686 | 0.522 |

## 4.3 Factors influencing the variability of TDA

Based on the observation in §4.2 that TDA values are often dominated by noise, we delve into the factors that lead to the instability of data attributions. We inspect the contribution of model initialisation, training set size and model complexity.

**Source of randomness.** From a Bayesian ML perspective, the stochasticity of TDA stems from the inherent uncertainty of the learned model posterior $p(\theta|\mathcal{D})$. We consider two sources of randomness, model initialisation (DE-Init) and SGD batch composition (DE-Batch). Results are reported in Table 1. For MNIST3 and CIFAR10, we observe that DE-Batch introduces lower levels of noise in the TDA estimates (lower p-values). Particularly on MNIST3, both LOO and other TDA methods result in statistically significant p-values ($< 0.005$). This implies that almost every training data $z_j$ is influencing every test data $z$ consistently across various batch compositions. We conclude that the greater source of variations for the TDA estimates is the model initialisation.

**Training set size.** We study how training set size is a source of noise (cf. Figure 4). We train the CNN2-L with different-size datasets of MNIST3 and CIFAR10, where we vary the number of samples per class. Batches are composed differently depending on the dataset size, meaning that parameter updates are made after processing different data. In addition, we train a CNN2-L on the complete MNIST dataset and use a subset of the training data for our experiments (cf. Appendix B.4). The results show a tendency for high variation in TDA scores with larger datasets first after which a decrease in variation is observed. The initial increase makes sense as the number of combinations for batch composition increases with dataset size. As the batching is initialised randomly during training, batches are likely to be composed of different data for larger datasets. This leads to variation in the learned model parameters, in turn affecting the reliability of TDA. At the point of decrease, the TDA scores are rather small for all train-test pairs. The attribution of individual training samples to a model prediction is overall small in larger datasets, which leads to a decrease in variance.

**Model complexity.** We study how model complexity is linked to the reliability of TDA estimates. See Table 2. We observe that, compared to the CNN models, a large ViT model trained with LoRA results in dramatically greater p-values. For example, for LOO on MNIST3, the p-value increases from 0.025 (CNN2-L) to 0.786. A similar trend is observed for other TDA methods. A less dramatic increase in p-values can also be observed with the addition of another layer to the CNN (i.e. from

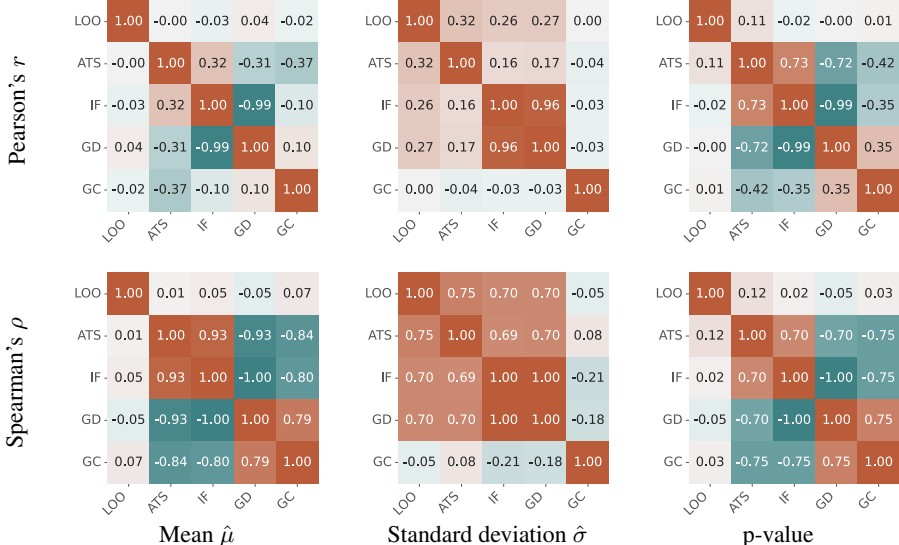

Figure 5: **Correlation of TDA methods.** Pearson and Spearman correlation coefficients among ground-truth TDA and approximate TDA methods. We show correlations for TDA mean $\hat{\mu}$, TDA standard deviation $\hat{\sigma}$, and TDA p-values. All results are based on the setting: CNN2-L, MNIST3, SWA+DE-Init.

CNN2-L to CNN3-L). This implies that the reliability of TDA estimates decreases with increasing model complexity. While we limit the number of trainable parameters in our ViT by using LoRA to be comparable to CNN3-L, the p-values computed from TDA estimates are significantly larger. Larger models exhibit a larger parameter space so that noise stemming from model initialisation or batch composition is amplified. While we fix the model initialisation and dataset size, the batch composition still varies across the model parameters $\theta$ sampled from the posterior $p(\theta|\mathcal{D})$ per model. As both the CNNs and ViT are trained with the same sampled batch compositions, we attribute the strong increase of p-value to the model complexity.

### 4.4 (Dis)agreement between TDA methods

We test the reliability of different TDA methods. Ideally, all methods approximate the ground-truth TDA (LOO). Yet the results suggest that there are substantial differences among the methods. For example, Grad-Cos is much more stable than all others. Hence, we study TDA methods with respect to both their correlation with LOO and among each other using Pearson and Spearman correlation of mean and variance of the TDA distributions, as proposed in §3.

Figure 5 shows the correlation matrices for one experiment (all experimental results in the Appendix). The results show that viewing TDA scores as distributions gives insights into the reliability of TDA methods: None of the tested TDA method's expected values $\hat{\mu}$ correlates with LOO. This implies that none of the TDA methods is a good approximation when the random factors are considered. The poor correlation of p-values of LOO indicates a disagreement in the train-test pairs considered low noise. We conclude that none of the tested methods reliably capture ground-truth TDA distributions.

Interestingly, we notice a stronger correlation between all other methods particularly when looking at the correlations of p-values. We identify two groups based on positive correlation, i.e. ATS with IF and GD with GC. Among the two groups, there is a negative correlation which indicates that methods interpret the sign of the attribution differently. Between IF, GD and GC this makes sense, as there is a negative sign in the definition of IF (Equation 2) which is not present in GD and GC. Considering absolute correlation, IF and GD are strongly correlated which shows that the dot product is a valid alternative for IF as they produce similar score distributions. The correlation between GD and GC indicates that the normalisation of gradients does not have a strong impact on the estimated TDA.

IF, GD and GC correlate considerably with ATS, which measures how the loss of a model on $z$ changes after doing one additional training step on $z_j$. Practically, ATS represents the gradient update after $z_j$, which is the same as the gradient $z_j$ itself. Therefore, it makes sense that the gradient-based approximation methods are close to ATS. We recognise a difference in the scope LOO and ATS

address. LOO looks at TDA globally and encapsulates the whole training, whereas ATS considers a local scope with a small model change. As IF, GD and GC correlate with ATS, we observe that they also correspond to a local change in the model, which underlines and extends the argument of [7]: There is a gap between LOO and IF, and more generally between the global and local view on TDA.

The TDA variance $\hat{\sigma}$ is noticeably well-correlated for TDA estimators and LOO, except for GC. This implies the existence of a consistent ranking of train-test pairs with stable attribution relationships. In particular, stable train-test pairs predicted by LOO are also likely to be stable pairs for TDA methods like ATS, IF, and GD. This motivates our final analysis and recommendation for evaluation in §4.5.

### 4.5 Considerations on TDA evaluation from a Bayesian perspective

Our analysis shows that both TDA estimates and ground-truth TDA values are affected by the noise stemming from the stochastic nature of deep model training. Hence, the practice of comparing against such a ground truth is destined to result in fragile estimates. We propose to treat TDA estimates as random variables which allows us to look at the evaluation from a Bayesian perspective: The comparison of TDA estimates against target TDA values is a comparison of two random variables. Since it is impossible to get rid of noise, it is better to compare distributions rather than point estimates. This provides an understanding of how well methods approximate the ground-truth distribution.

We observe that p-values vary between individual train-test sample pairs $(z_j, z)$; not all TDA estimates are equally affected by stochasticity. Interestingly, the presence of low-noise pairs is consistent across the majority of our experiments (cf. Figure 3), with varying sizes of the low-noise fraction. We find that fixing model initialisation and a small dataset size gives rise to a larger number of low-noise pairs. **We propose to focus on such low-noise pairs in TDA evaluation as their estimates are low in variance, leading to a more reliable evaluation.** Identifying such pairs requires an analysis similar to this work: treating TDA values as distributions and sampling multiple times from the posterior to get an estimate of the noise. It is crucial to find low-noise pairs to base evaluations on and understand when TDA is applicable. If no low-variance pairs exist, TDA cannot be used.

## 5 Related work

We study the reliability of TDA methods and add to the existing body of work on the fragility of TDA methods. Previous studies focused primarily on IF [29, 6, 8, 9, 7]. We extend the analysis by additionally studying other TDA methods. While IFs are theoretically grounded in robust statistics [15], they are based on two assumptions which are not always fulfilled in the context of deep learning: Twice-differentiability and strict convexity of the loss [3]. Zhang & Zhang [29] and Basu *et al.* [6] point to the fragility of the influence scores due to the non-convexity of deep learning. Particularly increasing model size is connected to increased model curvature, which means that influence estimates are more fragile with larger models. They find that strong regularisation is needed to improve estimation quality. Our experiments verify the observation that fragility increases with model size, which we observe across methods. We add that sources of randomness in the training process attribute to the fragility of TDA methods with increasing model size. Furthermore, related work found that the size of the training set contributes to the fragility of influence estimates. The attribution of one sample in a large training set is marginal so both influence estimates and ground-truth influence scores (i.e., from retraining the model) are noisy [6, 8, 9]. Through a Bayesian lens, we connect the increased fragility with increasing dataset size to batch composition as well. Not only is the attribution of a single sample in a large dataset marginal [6] but batches have vastly different compositions in larger datasets, introducing noise. A recent work [7] states that influence functions in deep learning do not correspond to LOO and quantify gaps in the estimation stemming from model non-linearity. A different approach in TDA [30, 31] aims at predicting the expected model output given a set of data points, directly considering randomness stemming from model initialisation. K & Søgaard [9] recommend reporting expected TDA scores to increase estimation stability. This approach is closest to our work but misses the consideration of variance in TDA estimates which we include by taking a Bayesian viewpoint.

In contrast to related work, we treat TDA values as distributions, which enables a novel perspective on the TDA task for deep models. We highlight the importance of considering the variance when studying reliability.

## 6  Conclusion

We adopt a Bayesian perspective on the training data attribution (TDA) methods to study their reliability when applied to deep models, given the stochastic nature of deep model training. By modelling TDA scores as distributions, we find that randomness in the training process, particularly due to parameter initialisation and batch composition, translates to variation in ground-truth TDA. We empirically observe that current estimation methods, such as influence functions, model a local change in the model whereas the ground truth attribution considers a global model change. Therefore, TDA is subject to inherent variance, leading us to suggest to the community: (1) When proposing a novel TDA method, one should view TDA from a Bayesian perspective and study the TDA estimates as distributions. (2) When using TDA, one should consider the variance to understand when the estimate can be trusted.

**Limitations.**   We perform an exhaustive analysis of TDA values $\tau$ and the estimates $\tau'$ for all train-test pairs $(z_j, z)$. Because of considerable computational costs, we have subsampled the datasets. In practice, datasets are considerably larger. Moreover, we choose simple tasks to eliminate the need for an additional hyperparameter search for model training, as the principal focus is on studying TDA methods. We choose gradient-based TDA methods but acknowledge that there exist many more, that we do not address. Hence, we encourage further study of TDA methods to fill these gaps and recommend investigating TDA from a Bayesian perspective, particularly in the low-data regime.

**Broader impact.**   This paper contributes to the field of data-driven XAI which aims at helping humans understand the inner workings of opaque models through data-centric approaches. Our work contributes to understanding the reliability of TDA methods and rethinking their evaluation against a noisy ground truth, which could help assess when TDA is appropriate and reliable.

## Acknowledgments and Disclosure of Funding

Kay Choi has helped in designing Figures 1 and 2. The authors thank the International Max Planck Research School for Intelligent Systems (IMPRS-IS) for supporting Elisa Nguyen. This work was supported by the Tübingen AI Center.

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

# A Model training details

We provide the source code at `https://github.com/ElisaNguyen/bayesian-tda`. All experiments were run on a single Nvidia 2080ti GPU.

## A.1 Data sampling

We use subsampled versions of the openly available MNIST [24] and CIFAR10 [25] datasets. For this, we first add an index which we use for randomly sampling a fixed number of images from each class. Table 3 includes the dataset sizes of the different experiments.

## A.2 Model training

CNN2-L has two convolutional layers followed by two fully connected linear layers, with GeLU activation. We use the Adam optimizer with a learning rate of 0.001 and a weight decay of 0.005. We use the cross-entropy loss and train the model for 15 epochs on MNIST3 and for 30 epochs on CIFAR10 with a batch size of 32. CNN3-L has 3 convolutional layers followed by two fully connected linear layers with GeLU activation. The hyperparameters are the same as for CNN2-L. For training the ViT with LoRA, we use the `peft` [32] and HuggingFace `transformers` library [33]. We start from the pretrained model checkpoint of [34] and finetune the LoRA model with the same hyperparameters as the CNN.

An overview of the predictive performance measured in accuracy on the subsampled training and test sets is provided in Table 3.

**Hint for reproducibility.** In particular, we use `CrossEntropyLoss(reduction='none')` during model training and also update this in the ViT training script `modeling_vit.py`. This is important for the LOO experiments, where we exclude a sample $z_j$ from contributing to the training by zeroing out the loss.

Table 3: Predictive performance at 95% CI across 10 runs (computed as 1.96*SE)

| Experiment | | | | | | |
|---|---|---|---|---|---|---|
| Model | Data | Randomness | $|\mathcal{D}_{\text{train}}|$ | $|\mathcal{D}_{\text{test}}|$ | Accuracy$_{\text{train}}$ | Accuracy$_{\text{test}}$ |
| CNN2-L | MNIST3 | SWA+DE-Init | 30 | 900 | 0.987±0.010 | 0.953±0.003 |
| CNN2-L | MNIST3 | SWA+DE-Init | 60 | 900 | 0.985±0.007 | 0.970±0.004 |
| CNN2-L | MNIST3 | SWA+DE-Init | 90 | 900 | 0.995±0.003 | 0.939±0.005 |
| CNN2-L | MNIST3 | SWA+DE-Init | 120 | 900 | 0.999±0.001 | 0.941±0.008 |
| CNN2-L | MNIST3 | SWA+DE-Init | 150 | 900 | 0.998±0.004 | 0.970±0.003 |
| CNN2-L | MNIST3 | SWA+DE-Init | 180 | 900 | 1.000 ±0.000 | 0.985±0.001 |
| CNN2-L | CIFAR10 | SWA+DE-Init | 100 | 500 | 0.989±0.020 | 0.260±0.010 |
| CNN2-L | CIFAR10 | SWA+DE-Init | 200 | 500 | 1.000±0.000 | 0.328±0.007 |
| CNN2-L | CIFAR10 | SWA+DE-Init | 300 | 500 | 1.000±0.000 | 0.360±0.005 |
| CNN2-L | CIFAR10 | SWA+DE-Init | 400 | 500 | 0.983±0.020 | 0.362±0.011 |
| CNN2-L | CIFAR10 | SWA+DE-Init | 500 | 500 | 0.992±0.014 | 0.377±0.012 |
| CNN2-L | CIFAR10 | SWA+DE-Init | 600 | 500 | 0.994±0.005 | 0.394±0.011 |
| CNN2-L | MNIST3 | SWA+DE-Batch | 150 | 900 | 0.993±0.002 | 0.975±0.002 |
| CNN2-L | CIFAR10 | SWA+DE-Batch | 500 | 500 | 0.991±0.010 | 0.364±0.003 |
| CNN3-L | MNIST3 | SWA+DE-Init | 150 | 900 | 0.994±0.005 | 0.971±0.008 |
| CNN3-L | CIFAR10 | SWA+DE-Init | 500 | 500 | 0.989±0.008 | 0.363±0.011 |
| ViT+LoRA | MNIST3 | SWA+DE-Batch | 150 | 900 | 0.945±0.008 | 0.935±0.005 |
| ViT+LoRA | CIFAR10 | SWA+DE-Batch | 500 | 500 | 0.934±0.006 | 0.892±0.008 |

# B Additional experimental results

We study the reliability of TDA estimates and values through a hypothesis test, where we report the p-values as an indication of the statistical significance of the TDA estimate. In this appendix, we

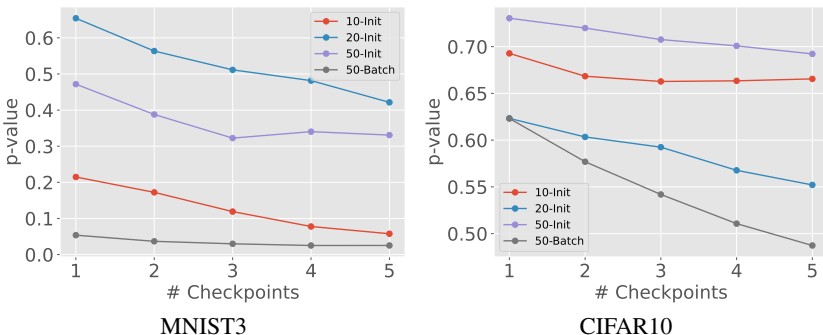

Figure 6: P-values of ground-truth TDA (LOO) with increasing number of SWA samples (i.e., number of model checkpoints used as samples of $\theta$) for the CNN trained on MNIST3 and CIFAR10.

provide a complete overview of the analyses of p-values and correlations between different TDA methods.

## B.1  DE vs. DE+SWA

In our work, we sample trained models $\theta$ sampled from the posterior $p(\theta|\mathcal{D})$. Concretely, we train the model across 10 different random seeds and record the checkpoints after each of the last five epochs of training. Each of these models represents a sample. We test how the stability of the TDA values $\tau(z_j, z)$ (LOO) behaves when we ensemble different numbers of checkpoints by investigating the mean p-values across all train-test pairs $(z_j, z)$. Figure 6 shows that higher numbers of samples $\theta$ increase stability, therefore we use all available samples in our subsequent analyses.

## B.2  Correlation analysis on low-noise pairs

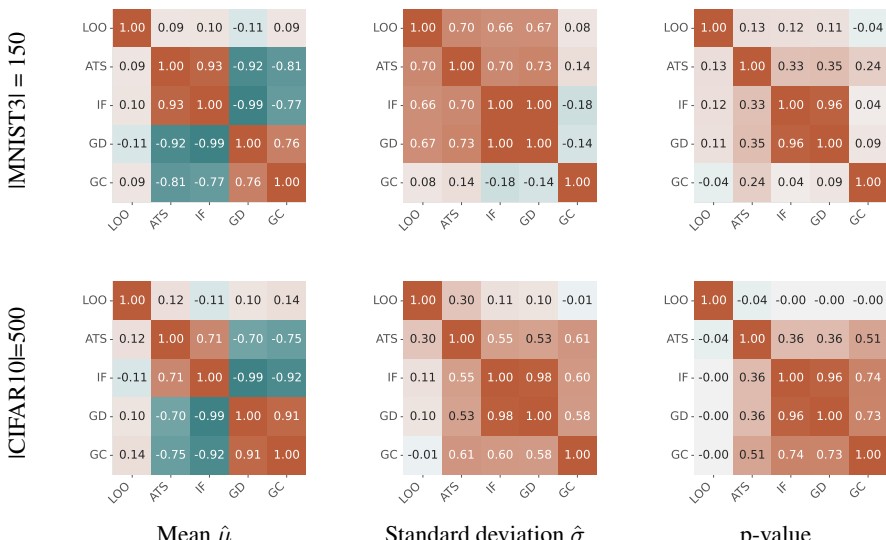

Figure 7: **Correlation analysis of low-noise train-test pairs** Spearman rank correlation coefficients for TDA scores with $p < 0.05$ of LOO retraining of the 2-layer CNN models trained on the respective datasets.

We analyse the Spearman correlation between the TDA approximation methods and ground-truth TDA (LOO) for low-noise train test pairs (i.e. where p-value of LOO scores is $\leq 0.05$). The correlation matrices are shown in Figure 7. Within the subset of low-noise train-test pairs, the Spearman correlation of the mean between LOO and the approximation methods is higher than when considering the whole dataset for both MNIST3 and CIFAR10. The correlation of $\hat{\sigma}$ of the

low-noise analysis is similar to the results found in the correlation analysis of the whole dataset (cf. Figures 12, 15). Therefore, we observe similar trends when considering only the low-noise train test pairs to the complete dataset: There is a weak correlation between the TDA approximation methods and LOO, while the TDA approximation methods correlate strongly among themselves. This reflects the difference between the global change of retraining and the local approximations.

## B.3   Mislabel identification

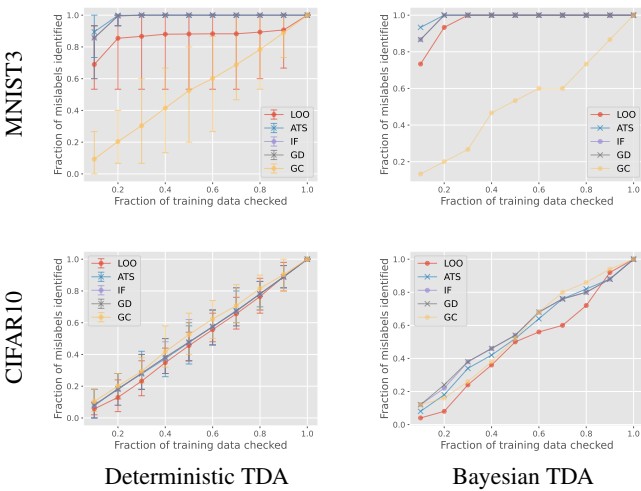

Figure 8: Fractions of mislabeled samples discovered with the deterministic definition of TDA (left, Equation 1) vs. a probabilistic definition of TDA (right, Equation 7).

A common way of evaluating TDA methods is through the auxiliary task of mislabel identification. We perform an this experiment with CNN2-L trained on MNIST3 ($|\mathcal{D}| = 150$) and CIFAR10 ($\mathcal{D} = 500$). We follow the procedure from Koh & Liang [3]: First, a random 10% of the datasets are mislabeled by the highest scoring incorrect label. We train the model using these mislabeled datasets (sample $T = 50$ times from the posterior). Then, we compute self-influence, which is the attribution of a sample to itself, with each TDA method. The mislabeled dataset is ranked according to self-influence and the quantity of interest is the fraction of mislabeled samples found when inspecting the top x% of the ranked dataset. In the analysis, we inspect (1) the range of mislabeled fractions discovered if we treat TDA deterministically, i.e. we compute the discovered fraction per $t \in T$ and report the range (cf. Figure 8 left); (2) the fraction of mislabeled samples discovered when we use the mean over the TDA scores of our posterior samples (cf. Figure 8 right). We find that deterministic TDA results in a large range of possible outcomes for identifying mislabeled samples. This means that it is harder to reliably identify mislabels when TDA is treated as point estimates.

## B.4   Experiments on MNIST (|D|=60,000)

Table 4: Mean p-values of ground-truth TDA (LOO) and estimated TDA values of with randomness included by SWA+DE-Init for CNN2-L trained on the complete MNIST dataset.

| LOO | ATS | IF | GD | GC |
|---|---|---|---|---|
| 0.761 | 0.362 | 0.464 | 0.475 | 0.247 |

In addition to our main experiments, we conduct a statistical test on the TDA scores obtained from a CNN2-L model trained on the complete MNIST ($|\mathcal{D}| = 60,000$) dataset (cf. Figure 4). We base the analysis on $T = 50$ samples from the model posterior (10 random seeds $\times$ 5 checkpoints) and a small random subsample of data (100 training samples $\times$ 10 test samples = 1000 train-test pairs) due to the high computational cost of retraining. We find high (p>0.05) variance in ground-truth TDA as well as TDA estimates, in line with findings from §4.3*Training set size*.

The results show that TDA estimates vary strongly for the small subset of train-test pairs (high p-values). We identify two main reasons: (1) MNIST is larger in training set size than the initial sets we used. The attribution of one sample to model behaviour is likely to be marginal and unstable. (2) Low-noise samples exist, but they are in the minority. This experiment shows that LOO is affected by the stochasticity of deep model training and TDA approximation methods fail to capture this.

## B.5   All results: Stability of TDA values and estimates

Table 5 presents the complete table of p-values of all tested TDA methods across all experiments of our work except for the experiments detailed in Appendices B.3 and B.4. Below, we display the histograms of p-values per experiment corresponding to each line in the table captioned with the experiment ID. The histograms show that low-noise train-test pairs $(z_j, z)$ are present in all experiments involving the CNN model (i.e., experiments 1-8), where the number of pairs varies. Generally, we observe that there is no connection between the size of the dataset and the distribution of p-values. Furthermore, we notice that fixing the model initialisation (i.e., randomness induced by SWA+DE-Batch) increases the number of stable train-test pairs (cf. experiment 5 to 13, 11 to 14). However, in the case of the ViT experiments, stable train-test pairs are practically non-existent which shows that model complexity affects the stability of TDA.

Table 5: Complete list of mean p-values of TDA values for all experiments.

| | | Experiment | | | | | | | |
|---|---|---|---|---|---|---|---|---|---|
| ID | Model | Data | Randomness | $|\mathcal{D}_{\text{train}}|$ | LOO | ATS | IF | GD | GC |
| 1 | CNN2-L | MNIST3 | SWA+DE-Init | 30 | 0.058 | 0.088 | 0.111 | 0.110 | 0.020 |
| 2 | CNN2-L | MNIST3 | SWA+DE-Init | 60 | 0.421 | 0.148 | 0.251 | 0.253 | 0.002 |
| 3 | CNN2-L | MNIST3 | SWA+DE-Init | 90 | 0.714 | 0.355 | 0.466 | 0.470 | 0.209 |
| 4 | CNN2-L | MNIST3 | SWA+DE-Init | 120 | 0.675 | 0.346 | 0.469 | 0.472 | 0.218 |
| 5 | CNN2-L | MNIST3 | SWA+DE-Init | 150 | 0.331 | 0.254 | 0.352 | 0.363 | 0.003 |
| 6 | CNN2-L | MNIST3 | SWA+DE-Init | 180 | 0.374 | 0.254 | 0.355 | 0.356 | 0.001 |
| 7 | CNN2-L | CIFAR10 | SWA+DE-Init | 100 | 0.665 | 0.424 | 0.607 | 0.608 | 0.352 |
| 8 | CNN2-L | CIFAR10 | SWA+DE-Init | 200 | 0.552 | 0.397 | 0.450 | 0.452 | 0.399 |
| 9 | CNN2-L | CIFAR10 | SWA+DE-Init | 300 | 0.543 | 0.389 | 0.456 | 0.456 | 0.313 |
| 10 | CNN2-L | CIFAR10 | SWA+DE-Init | 400 | 0.619 | 0.418 | 0.562 | 0.568 | 0.344 |
| 11 | CNN2-L | CIFAR10 | SWA+DE-Init | 500 | 0.692 | 0.438 | 0.575 | 0.587 | 0.356 |
| 12 | CNN2-L | CIFAR10 | SWA+DE-Init | 600 | 0.665 | 0.447 | 0.575 | 0.579 | 0.358 |
| 13 | CNN2-L | MNIST3 | SWA+DE-Batch | 150 | 0.025 | 0.039 | 0.000 | 0.000 | 0.000 |
| 14 | CNN2-L | CIFAR10 | SWA+DE-Batch | 500 | 0.623 | 0.374 | 0.535 | 0.534 | 0.314 |
| 15 | CNN3-L | MNIST3 | SWA+DE-Init | 150 | 0.370 | 0.368 | 0.464 | 0.479 | 0.005 |
| 16 | CNN3-L | CIFAR10 | SWA+DE-Init | 500 | 0.687 | 0.432 | 0.579 | 0.581 | 0.365 |
| 17 | ViT+LoRA | MNIST3 | SWA+DE-Batch | 150 | 0.786 | 0.573 | 0.369 | 0.365 | 0.093 |
| 18 | ViT+LoRA | CIFAR10 | SWA+DE-Batch | 500 | 0.777 | 0.766 | 0.686 | 0.686 | 0.522 |

## B.6   All results: Correlation analysis

In the main body of this paper, we report the Pearson and Spearman correlation matrices for experiment 3 (CNN2-L trained on MNIST3 with 50 samples per class and randomness induced by SWA+DE-Init). This section presents the complete overview of correlations between TDA methods across all experiments in Figures 10 - 27. We note that the observations and analyses in the main paper hold across experiments.

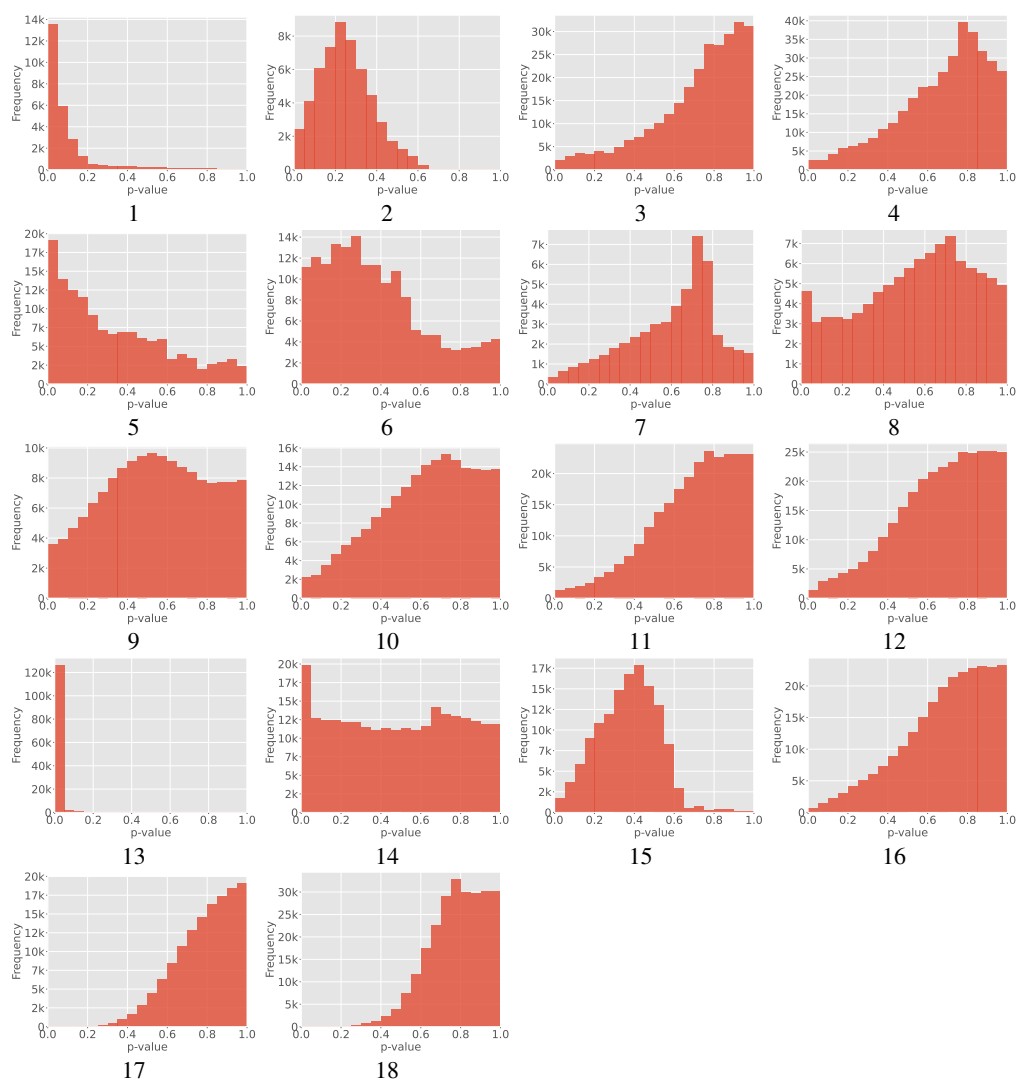

Figure 9: Distribution of p-values for ground-truth TDA (LOO) for all experiments (IDs corresponding to IDs in Table 5).

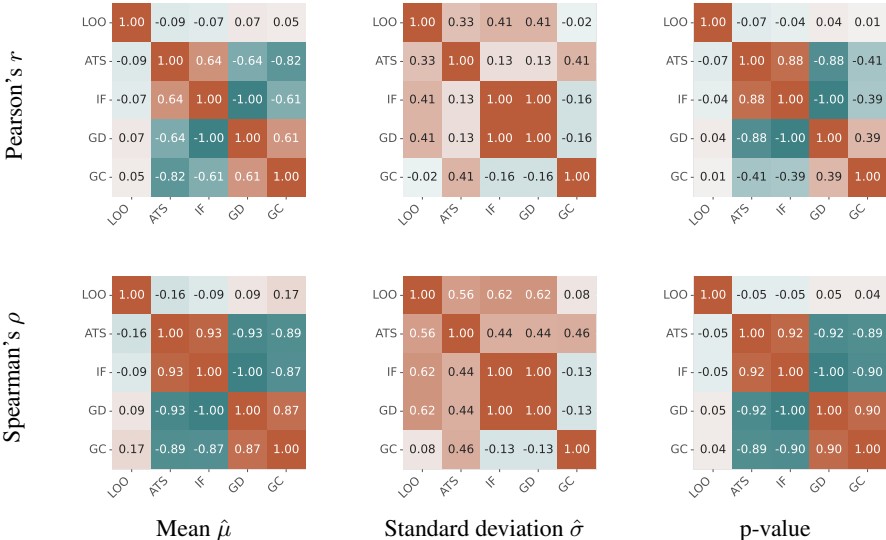

Figure 10: **Experiment 1 (cf. Table 5):** Pearson and Spearman correlation coefficients among ground-truth TDA and approximate TDA methods. We show correlations for TDA mean $\hat{\mu}$, TDA standard deviation $\hat{\sigma}$, and TDA p-values.

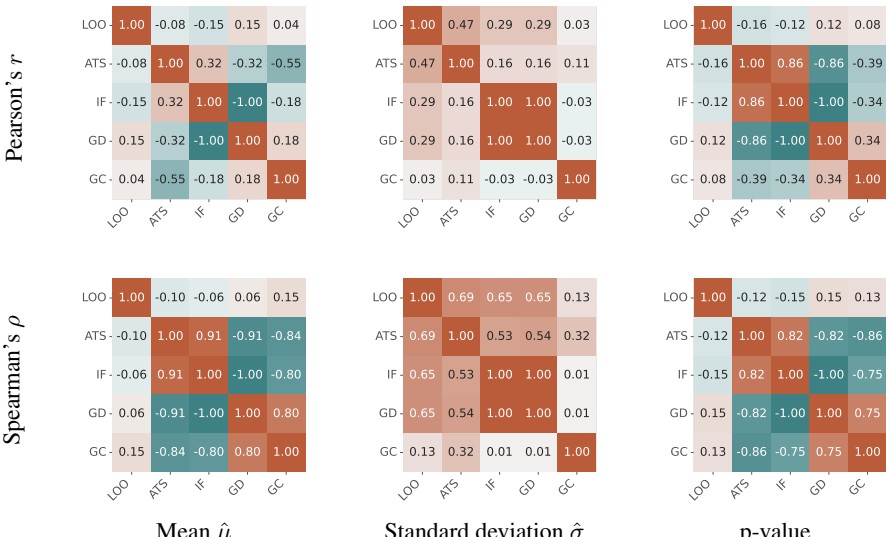

Figure 11: **Experiment 2 (cf. Table 5):** Pearson and Spearman correlation coefficients among ground-truth TDA and approximate TDA methods. We show correlations for TDA mean $\hat{\mu}$, TDA standard deviation $\hat{\sigma}$, and TDA p-values.

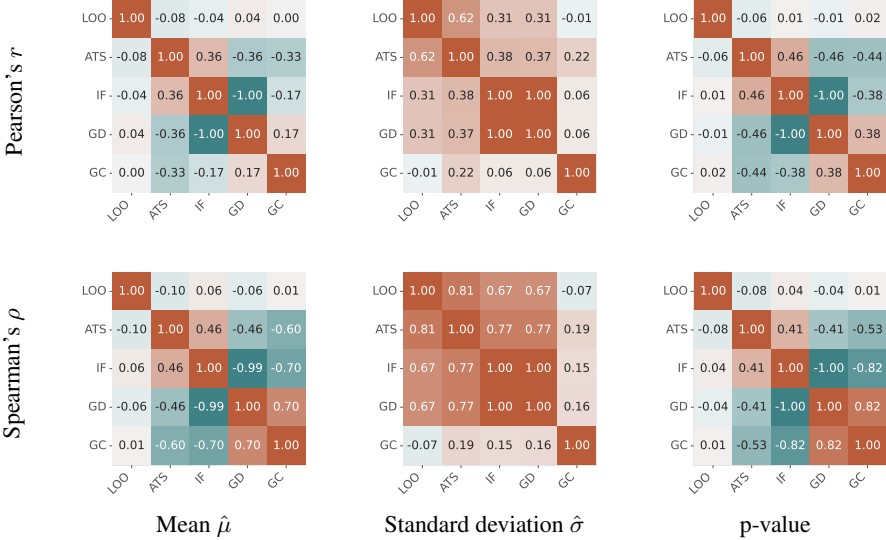

Figure 12: **Experiment 3 (cf. Table 5):** Pearson and Spearman correlation coefficients among ground-truth TDA and approximate TDA methods. We show correlations for TDA mean $\hat{\mu}$, TDA standard deviation $\hat{\sigma}$, and TDA p-values.

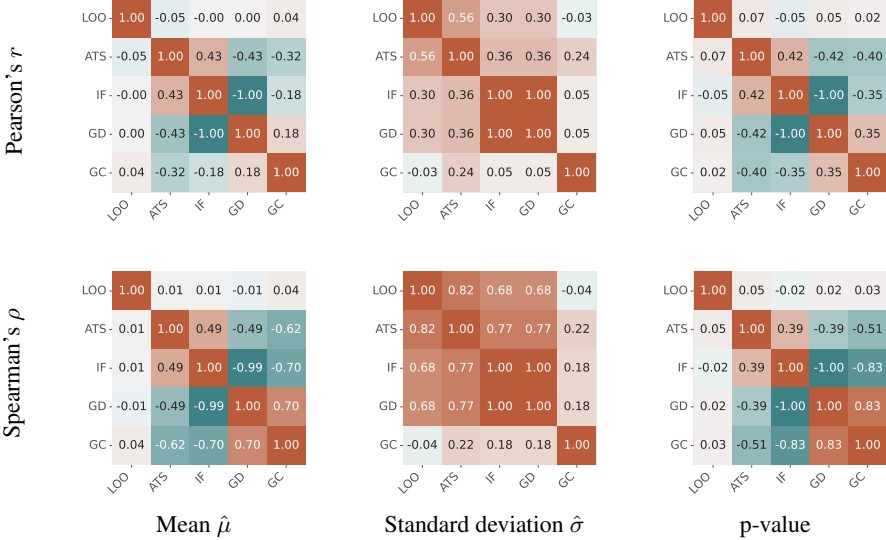

Figure 13: **Experiment 4 (cf. Table 5):** Pearson and Spearman correlation coefficients among ground-truth TDA and approximate TDA methods. We show correlations for TDA mean $\hat{\mu}$, TDA standard deviation $\hat{\sigma}$, and TDA p-values.

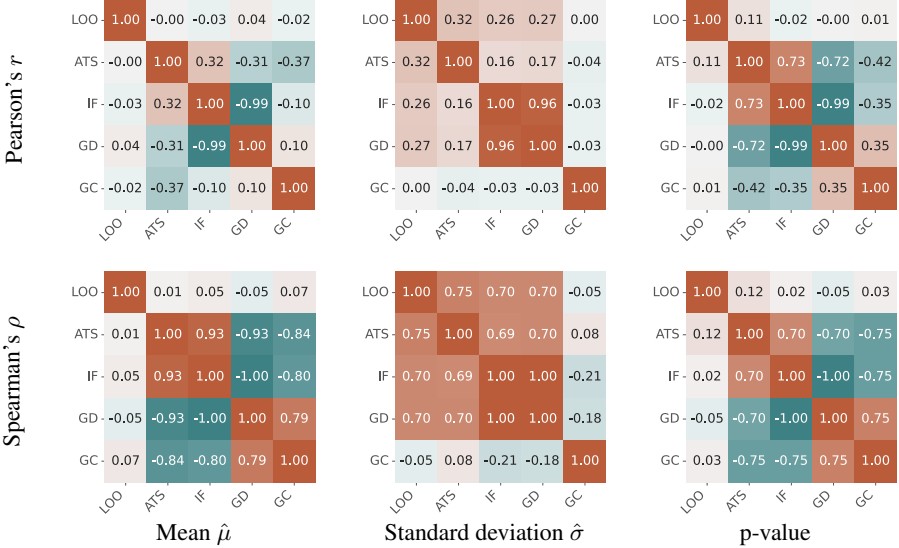

Figure 14: **Experiment 5 (cf. Table 5):** Pearson and Spearman correlation coefficients among ground-truth TDA and approximate TDA methods. We show correlations for TDA mean $\hat{\mu}$, TDA standard deviation $\hat{\sigma}$, and TDA p-values.

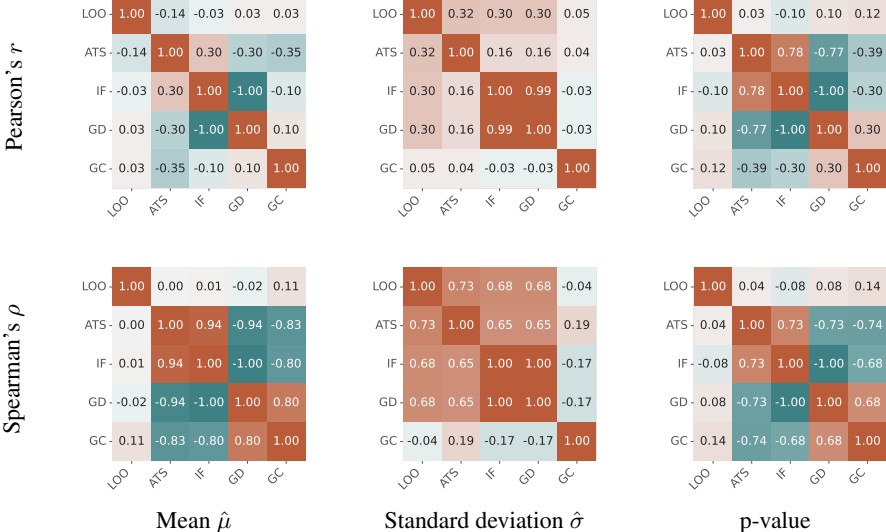

Figure 15: **Experiment 6 (cf. Table 5):** Pearson and Spearman correlation coefficients among ground-truth TDA and approximate TDA methods. We show correlations for TDA mean $\hat{\mu}$, TDA standard deviation $\hat{\sigma}$, and TDA p-values.

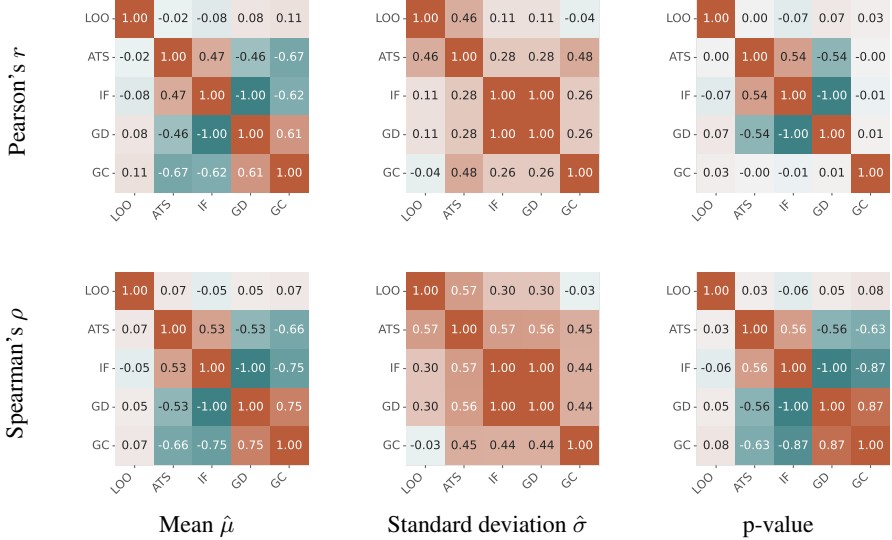

Figure 16: **Experiment 7 (cf. Table 5):** Pearson and Spearman correlation coefficients among ground-truth TDA and approximate TDA methods. We show correlations for TDA mean $\hat{\mu}$, TDA standard deviation $\hat{\sigma}$, and TDA p-values.

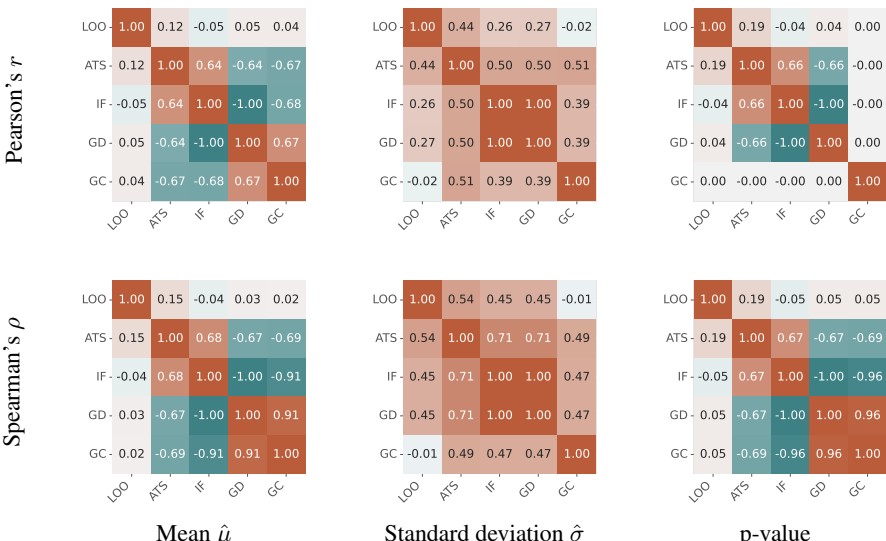

Figure 17: **Experiment 8 (cf. Table 5):** Pearson and Spearman correlation coefficients among ground-truth TDA and approximate TDA methods. We show correlations for TDA mean $\hat{\mu}$, TDA standard deviation $\hat{\sigma}$, and TDA p-values.

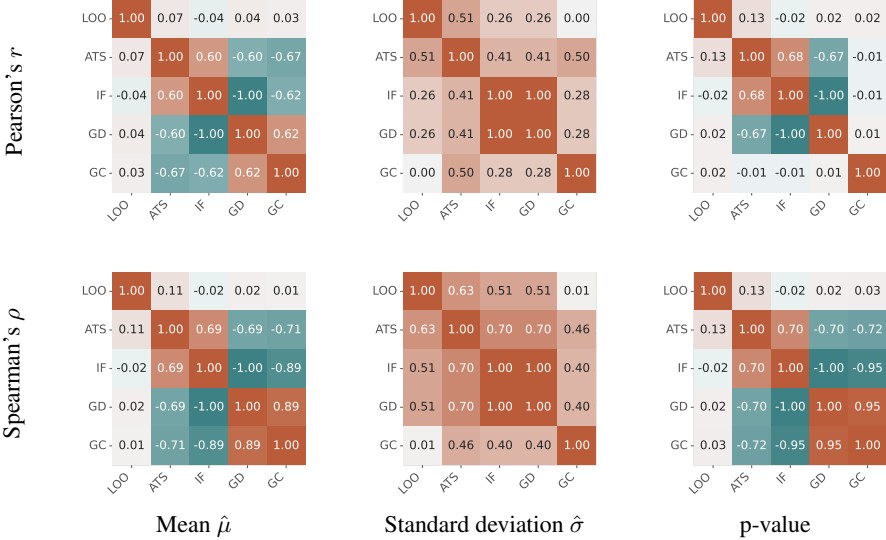

Figure 18: **Experiment 9 (cf. Table 5):** Pearson and Spearman correlation coefficients among ground-truth TDA and approximate TDA methods. We show correlations for TDA mean $\hat{\mu}$, TDA standard deviation $\hat{\sigma}$, and TDA p-values.

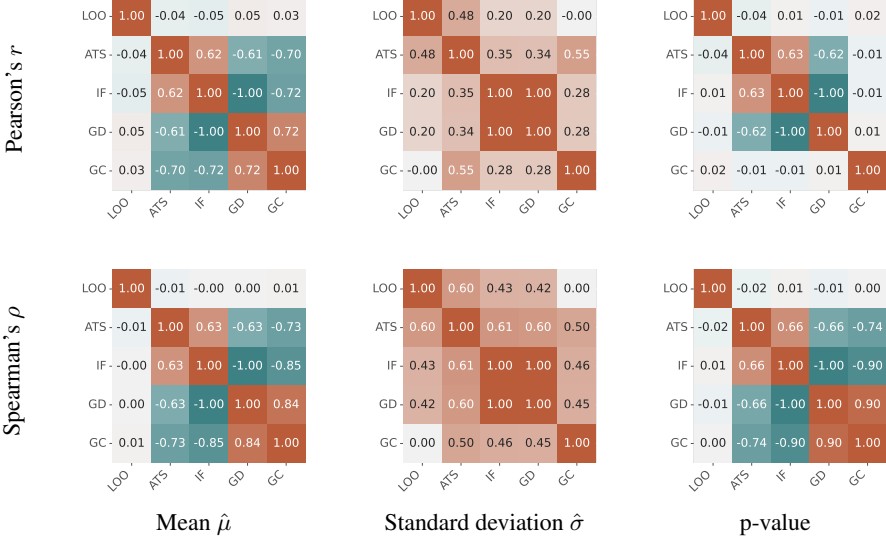

Figure 19: **Experiment 10 (cf. Table 5):** Pearson and Spearman correlation coefficients among ground-truth TDA and approximate TDA methods. We show correlations for TDA mean $\hat{\mu}$, TDA standard deviation $\hat{\sigma}$, and TDA p-values.

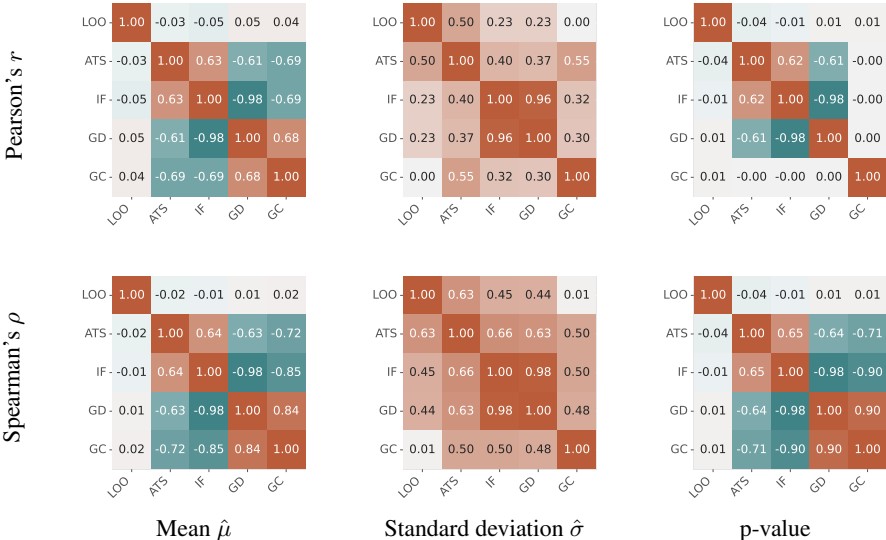

Figure 20: **Experiment 11 (cf. Table 5):** Pearson and Spearman correlation coefficients among ground-truth TDA and approximate TDA methods. We show correlations for TDA mean $\hat{\mu}$, TDA standard deviation $\hat{\sigma}$, and TDA p-values.

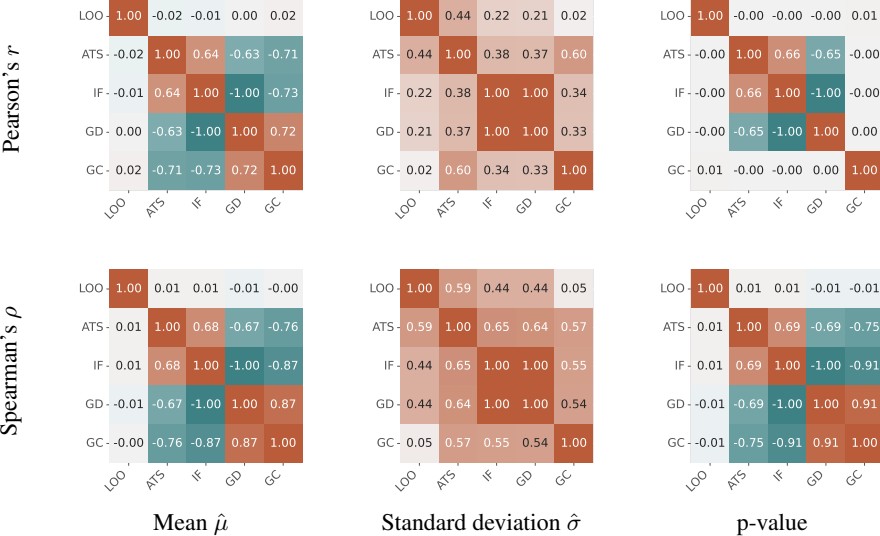

Figure 21: **Experiment 12 (cf. Table 5):** Pearson and Spearman correlation coefficients among ground-truth TDA and approximate TDA methods. We show correlations for TDA mean $\hat{\mu}$, TDA standard deviation $\hat{\sigma}$, and TDA p-values.

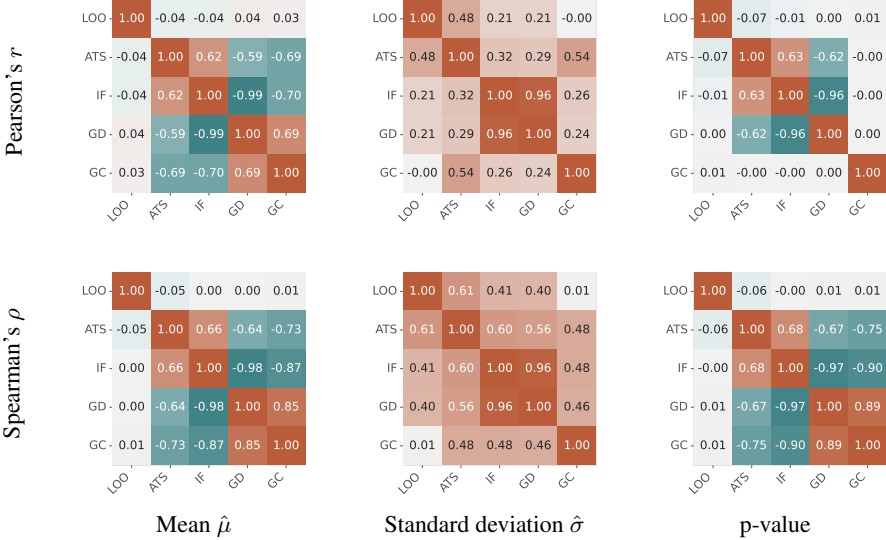

Figure 22: **Experiment 13 (cf. Table 5):** Pearson and Spearman correlation coefficients among ground-truth TDA and approximate TDA methods. We show correlations for TDA mean $\hat{\mu}$, TDA standard deviation $\hat{\sigma}$, and TDA p-values.

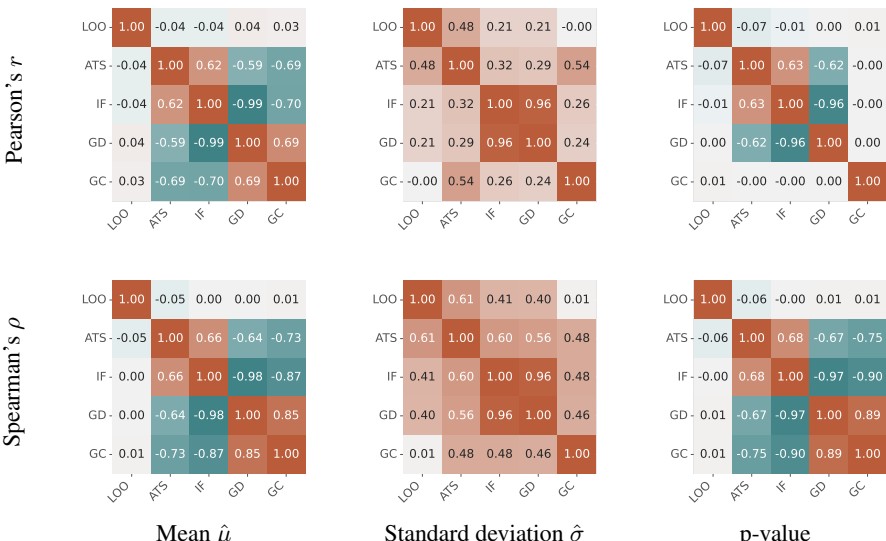

Figure 23: **Experiment 14 (cf. Table 5):** Pearson and Spearman correlation coefficients among ground-truth TDA and approximate TDA methods. We show correlations for TDA mean $\hat{\mu}$, TDA standard deviation $\hat{\sigma}$, and TDA p-values.

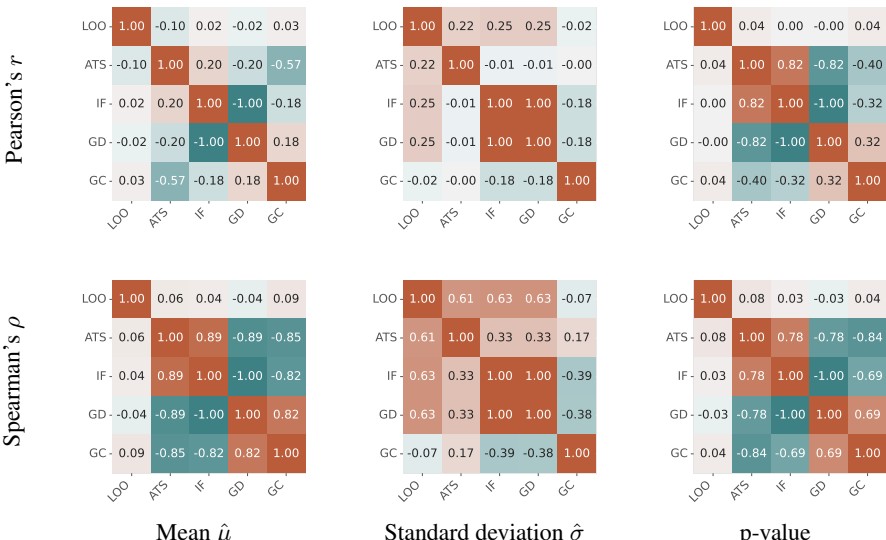

Figure 24: **Experiment 15 (cf. Table 5):** Pearson and Spearman correlation coefficients among ground-truth TDA and approximate TDA methods. We show correlations for TDA mean $\hat{\mu}$, TDA standard deviation $\hat{\sigma}$, and TDA p-values.

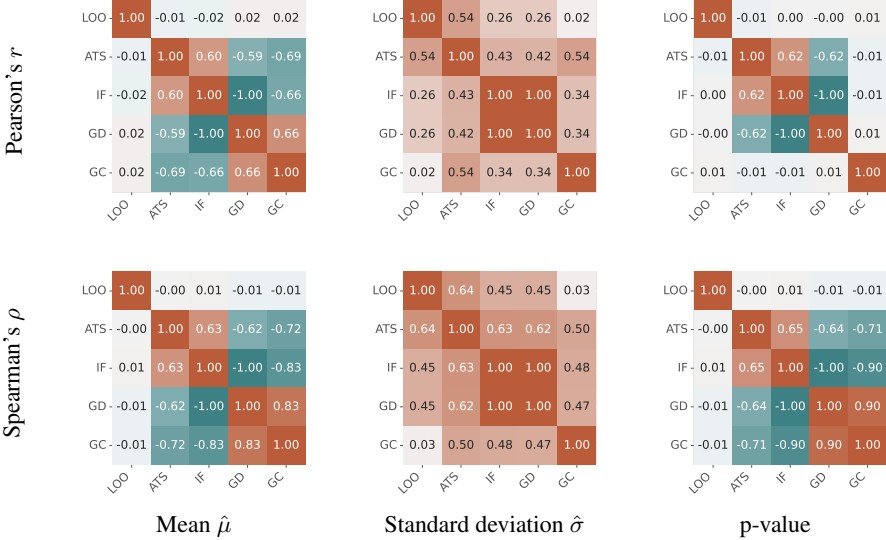

Figure 25: **Experiment 16 (cf. Table 5):** Pearson and Spearman correlation coefficients among ground-truth TDA and approximate TDA methods. We show correlations for TDA mean $\hat{\mu}$, TDA standard deviation $\hat{\sigma}$, and TDA p-values.

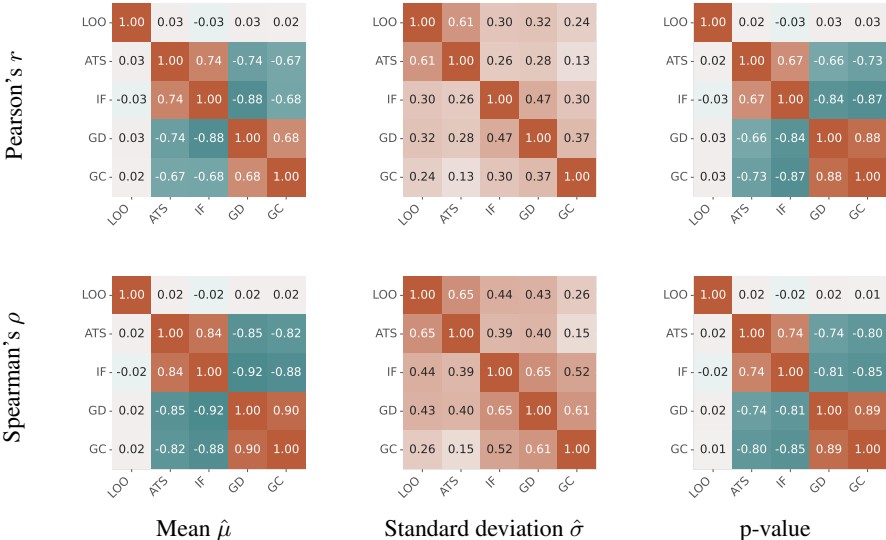

Figure 26: **Experiment 17 (cf. Table 5):** Pearson and Spearman correlation coefficients among ground-truth TDA and approximate TDA methods. We show correlations for TDA mean $\hat{\mu}$, TDA standard deviation $\hat{\sigma}$, and TDA p-values.

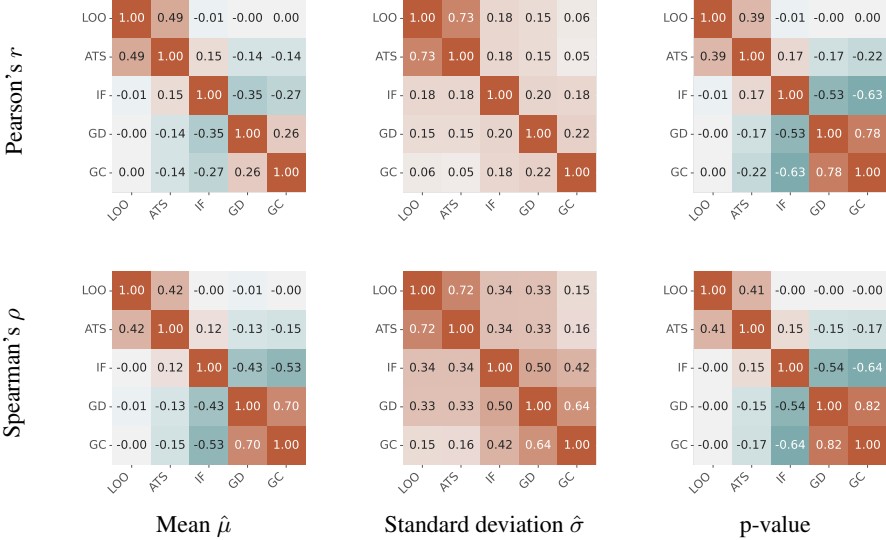

Figure 27: **Experiment 18 (cf. Table 5):** Pearson and Spearman correlation coefficients among ground-truth TDA and approximate TDA methods. We show correlations for TDA mean $\hat{\mu}$, TDA standard deviation $\hat{\sigma}$, and TDA p-values.

