# OpenReview forum: "A Bayesian Approach To Analysing Training Data Attribution In Deep Learning"
_NeurIPS.cc/2023/Conference — NeurIPS 2023 poster_

### Official Review · Reviewer_wqES · 2023-06-28

**Soundness:** 3 good
**Presentation:** 3 good
**Contribution:** 2 fair
**Rating:** 5
**Confidence:** 3

**Summary:**

The paper studies the challenges in measuring the performance of training data attribution (TDA) methods arising from stochasticity in training large deep neural networks. Specifically, the authors use various existing approaches to obtain many samples from the posteriors of the model weights instead of a point estimate with and without a training instance whose influence needs to be computed. The authors then compute the mean and variance of ground truth attribution/influence scores; and measure the performance of various attribution methods by looking at the correlation of these mean and variance with the ground truth scores.


**Strengths:**

- Thorough experiments comparing various Leave-One-Out (LOO) or attribution approximation methods including influence scores and variants (GD, GS), additional training step (ATS).
- The paper provides further empirical evidence to previous observations [1] that common approximations used for LOO actually estimate slightly different objects and are susceptible to randomness in training arising from initialization of model weights, batch order, etc.

**Weaknesses:**

- It is not clear if the phrase Bayesian perspective on TDA is useful, authors could say more on the connection between using Bayesian DL methods, Student t-test for measuring the noise in TDA estimates and Bayesian perspective on TDA.
- In eq 8, how are the T samples from the two posteriors (with and without a training instance) paired.
- It is also not clear if the observations made here only apply to measuring the attribution performance wrt to the ground truth attributions. That is, the main recommendation to focus on low signal to noise pairs seemly to apply only to evaluation. Did the authors perform such an evaluation? How does this recommendation affect the use of influence scores downstream applications such as correcting label mistakes, removing biased instances, etc.

[1] Bae, Juhan, Nathan Ng, Alston Lo, Marzyeh Ghassemi, and Roger B. Grosse. "If Influence Functions are the Answer, Then What is the Question?." Advances in Neural Information Processing Systems 35 (2022): 17953-17967.

**Questions:**

See Weaknesses section.

**Limitations:**

Yes, limitations are discussed.

---

> ### Author Rebuttal · Authors · 2023-08-10
>
> We thank the reviewer for their comments and questions. We hope to clarify concerns in the following.
>
> ### It is not clear if the phrase Bayesian perspective on TDA is useful, authors could say more on the connection between using Bayesian DL methods, Student t-test for measuring the noise in TDA estimates and Bayesian perspective on TDA.
> TDA estimates are usually treated as deterministic point estimates, even though model training is in fact probabilistic (SGD and model initialisation are probabilistic). Bayesian ML is a framework for modelling the uncertainty in model training (cf. Section 2.2 of the submission and [a]). In our work, we borrow methods from Bayesian ML [b,c] to turn TDA estimates into random variables. The random variables are in turn studied through the proposed statistical testing. By treating TDA estimates as a probabilistic distribution, we can examine the statistical significance of TDA estimates and quantify the variance which we found to be inherent in the TDA task.
>
> ### In eq 8, how are the T samples from the two posteriors (with and without a training instance) paired.
> We don't pair the samples for the two posteriors, as you can see in Eq (9):
> $$
> \text{Var} [\tau (z_j, z)] = \frac{1}{T^2}\sum_{t,t^\prime} \left(\mathcal{L}(z ;\theta^{(t)}_{\setminus j}) - \mathcal{L}(z ;\theta^{(t^\prime)})-\mathbb{E} [\tau (z_j, z)]\right)^2
> $$
>
> The reason we use common index $t$ for the two distributions in Eq (8) is because it results in the identical output due to the linearity:
>
> $$
> \mathbb{E} [\tau (z_j, z)] = \frac{1}{T}\sum_t \mathcal{L}(z ;\theta^{(t)}\_{\setminus j}) - \frac{1}{T}\sum\_{t^\prime} \mathcal{L}(z;\theta^{(t^\prime)}) = \frac{1}{T}\sum_t \mathcal{L}(z ;\theta^{(t)}_{\setminus j}) - \mathcal{L}(z ;\theta^{(t)})
> $$
>
> We hope this brought some clarity and are happy to discuss any follow-up questions further.
>
> ###  It is also not clear if the observations made here only apply to measuring the attribution performance wrt to the ground truth attributions. That is, the main recommendation to focus on low signal to noise pairs seemly to apply only to evaluation. Did the authors perform such an evaluation?
> No, we did not make an analysis of the set of low-noise pairs in the submission but it is an interesting point. The Spearman rank correlation matrices of the means, standard deviations and p-values for the set of low-noise train-test pairs (LOO p-values <0.05) are provided in the global response PDF for two experiments (MNIST3 $|\mathcal{D}|=150$ and CIFAR10 $|\mathcal{D}|=500$ with CNN).
> We can see from this analysis:
> The approximate TDA methods do not correlate strongly with LOO.
> In this subset, p-values correlate positively among approximate methods, underlining observations from Section 4.4 (Dis)Agreement of TDA methods: TDA estimation methods discover similar attribution including the stochastic noise.
> In this subset, means of TDA methods correlate stronger with LOO than for all train-test pairs (e.g. for MNIST3: ATS: 0.01 -> 0.09, IF: 0.05 ->0.10, GD: -0.05 -> -0.11, GC: 0.07 -> 0.09 (Fig 5, lower left matrix -> Rebuttal PDF file, upper left)
> This evaluation shows that none of the tested TDA methods approximate the ranking of the true change in loss, even if this change is stable wrt training process stochasticity. In the case of IF, the possible reason could lie in observations made by [d]: IFs do not correspond to pure LOO retraining, approximation gaps and solver errors lead to a different objective. A study regarding how the TDA estimations methods correspond to [d]’s objective would be interesting in the future.
>
> ### How does this recommendation affect the use of influence scores downstream applications such as correcting label mistakes, removing biased instances, etc.
> The question of how variance in TDA estimates may affect downstream tasks is intriguing. We ran an additional experiment for the downstream task of mislabel identification similar to [e] and [f] with MNIST3. We find that the inherent stochasticity of TDA leads to a large range in mislabel identification performance. High variance in the TDA estimates degrades downstream task performance. Further details in the global response, experiment 2.
>
> [a] Bayesian methods in global optimization (1991). \
> [b] If Influence Functions are the Answer, Then What is the Question? (NeurIPS 2022).\
> [c] Simple and scalable predictive uncertainty estimation using deep ensembles (NeurIPS 2017).\
> [d] Averaging weights leads to wider optima and better generalization (UAI 2018).\
> [e] Understanding black-box predictions via influence functions (ICML 2017).\
> [f] Estimating training data influence by tracing gradient descent (NeurIPS 2020).

---

> > ### Comment · Reviewer_wqES · 2023-08-17
> > **Thanks for the rebuttal**
> >
> > I would like to thank the authors for additional experiments and responding to my questions.

---

### Official Review · Reviewer_fqXw · 2023-07-04

**Soundness:** 2 fair
**Presentation:** 2 fair
**Contribution:** 2 fair
**Rating:** 4
**Confidence:** 4

**Summary:**

This paper aims to examine training data attribution (TDA) methods from a Bayesian perspective, assuming the learned model parameters are samples from a posterior distribution. The paper illustrates how this perspective might affect TDA methods. It conducts experiments comparing and contrasting different TDA methods when run several times on model parameter samples from an approximate posterior. Experiments consider subsampled versions of MNIST and CIFAR10, and examine two model classes CNN and ViT+LoRA.


**Strengths:**

The topic is very interesting.
As a researcher and practitioner that has worked extensively with TDA methods, I have found myself thinking about (and navigating) the relationship between randomness in model parameters and TDA methods.
I think the community stands to benefit from this line of investigation.
The paper communicates many of its ideas quite clearly. Figures 1 and 2 are especially helpful.
At a high level, the experimental setup is a sensible approach.
Anonymized source code is included.


**Weaknesses:**

The paper is lacking in its theoretical treatment of the subject it aims to explore (effectively limited to Figure 1). This reduces its value to the community. Conclusions are drawn based on experiments conducted via (relatively small) samples from a few simple approximations of the posterior. The paper would benefit considerably from some mathematical analysis and discussion of how a Bayesian model or perspective might affect the validity of TDA methods, such as influence functions. For example, how does treating $\theta$ as a random variable affect the application of the implicit function theorem used to derive influence? The Hessian of the loss is a function of $\theta$, for some set of values in the support of $\theta$ this Hessian is not invertible. Does this set have zero measure? Can it be safely ignored? It seems that there are numerous details that go unexamined.

Overall the writing is quite clear. But crucially, I found the presentation of the test statistic (Equation 11) to be confusing. As such it was difficult to interpret (and thus to review) the subsequent experimental results. (See questions below. I continue the review assuming that the p-values quantify the probability that the samples of $\tau$ could be generated by a random variable having mean equal to 0.)

At a high level, the experimental setup makes sense to me. However, some of the analytical steps in processing the collected data seem weak. For instance, on line 160 it is stated that “[this] treatment [...] poses a novel challenge for evaluation”. Yet if I understood correctly, the experiments yield samples from random variables that are compared in pairs, $\tau$ and $\tau’$. There are numerous established ways to compare two sample-sets, and to quantify the probability that they come from the same underlying distribution.  A two sample t-test may be an option.

Without much theoretical analysis to rest on, the experiments on two small datasets and two models struggle to convince me of the general claims. For example, the experiments on training set size (Section involve only three size settings, and don’t show clear trends. While the discussion on model complexity only considers two models which differ in both size and architecture.

Influence functions have been successfully used to reshape model behavior through the removal of identified sets of training instances. Several of the works cited in this paper achieve this. Therefore, the IF approximation must correlate with LOO at least on average for some types of training instances. How can this be reconciled with the results (e.g. in Figure 5)? This is not really discussed. Can they be explained via low noise pairs?

Minor:

Line 81: While Koh & Liang showed that IFs could be applied to neural networks, they did not derive the method. I suggest citing the original robust statistics papers on IF and/or the infinitesimal jackknife.

Figures 3 and 4 would benefit from larger labels.


**Questions:**

Regarding the presentation of the statistical test:

In Equation 7 $\tau$ is redefined as a random variable. The randomness stems from $\theta$ conditioned on the observed dataset and the removal of j. (It might be helpful to indicate that with the notation.)

In Equation 10, it states that the hypotheses being tested are whether \tau does or does not equal 0. This is unclear. I assume these hypotheses refer to $\tau$’s mean, and that the t-test is intended to determine the probability that the samples of $\tau$ could be generated by a random variable having mean equal to 0. Is this correct?

In Equation 11, what version of the T-test are you using? It would be helpful to cite your statistical method. Is the test designed to consider the pairing of the posterior (and perturbed posterior) samples?

In Equation 11, I assume that t should be indexed by j (and is also a function of the test sample z), but it’s unclear whether $t$ is computed for every pair of posterior (and perturbed posterior) samples. The left-most term of the numerator suggests this. If so, how are you aggregating these sets of values to get a single value per train-test pair?

Other:
Is it common to combine these approximate posterior sampling techniques? What are the implications of doing so?


**Limitations:**

The experiments in this paper consider two models. The paper notes limitations in the size of the datasets considered, explaining that it opted to spend computation time/budget on an exhaustive analysis of TDA values instead. However, if the objective was to comment on issues with TDA in practice, this choice (and limitation) is considerable. The results would be more convincing if the experiments had focused on only a subset of the test-train pairs, but considered larger (more realistic) dataset sizes, and a broader set of model architectures and/or sizes.

---

> ### Author Rebuttal · Authors · 2023-08-10
>
> Thank you for the thoughtful review and constructive suggestions.
> ### The paper [...] would benefit [...] from some mathematical analysis [...] of how a Bayesian [...] perspective might affect the validity of TDA.
> Our probabilistic conversion of TDA methods does not affect theoretical soundness. We apply the original TDA algorithms on the Monte-Carlo samples of model posteriors without modification. Since the methods are identical, it is unclear which additional theoretical treatment would add value. We are happy to discuss should there be further requests.
> ### Conclusions are drawn based on experiments [...] from a few simple approximations of the posterior.
> The simplicity of posteriors is the driving force behind the successful application of Bayesian ML on complex deep learning models [a,b,c,d]. While simple posteriors like Deep Ensemble (DE) [11] and Stochastic Weight Averging (SWA) [12] do not restore the posterior perfectly, they are sufficient to support our main message: TDA is inherently stochastic. Also, we highlight that we use T=50 posterior samples. This is not a small scale in both Bayesian ML and TDA communities. DE [11] and SWA [21] used T=15 and 30, resp. Previous work on TDA has used only a few deterministic models (T=1 [e], 3 [f]).
> ### How does treating θ as a random variable affect the [...] implicit function theorem [(IFT)]?
> Even in the deterministic setup, the IFT and inverse Hessians are generally inapplicable due to the non-convexity and high dimensionality of the deep learning optimisation problem. They are not novel challenges introduced by our probabilistic treatment. This is why Koh & Liang have proposed in [3] Section 4.2 (Non-convexity and non-convergence) to add a damping term λ to ensure the positive definiteness (PD) of Hessians. We used λ=3e-5 (as in [f]) and this guarantees PD Hessians at all posterior samples.
> ### Analytical steps [...] seem weak. [...] There are [many] established ways to compare two sample-sets
> Yes, there are many established ways to compare two random variables. However, we address a more complex problem. We need the rank correlation between two sets of random variables: $(\tau_1, …, \tau_N)$ vs $(\tau_1’, …, \tau_N’)$. $\tau_i$ refers to the TDA value of the $i^\text{th}$ train-test sample pair and $\tau_i’$ refers to the estimated TDA value.
> We are not aware of an existing tool for this problem. We have proposed to measure the Spearman’s ρ between the means and variances separately (Sec. 3), with the intuition that $\tau’$ must also faithfully rank the train-test pairs according to the level of noise.
> ### The experiments on two small datasets and two models struggle to convince me of the general claims.
> We agree that testing on larger datasets makes our claim stronger. We were limited by computational resources (e.g. LOO for CIFAR10 of size 500 requires ~104 GPU hours). Nonetheless, the experiments support our general claim: TDA is stochastic and evaluation protocols must reflect this. Besides, TDA tends to be more stable with smaller models. We expect that noise will dominate the signal in larger settings, making the analysis less meaningful.
> ### [Training set size experiments] involve only three size settings, and don’t show clear trends
> We ran additional experiments further varying the training set size. The results show that the variance in TDA estimates remains high after certain training set size. Exp. 3 in global response.
> ### [...] discussion on model complexity only considers two models which differ in both size and architecture.
> To address the reviewer’s concern that our models are too different and few, we ran an experiment with a 3-layer CNN that matches ViT+LoRA in #trainable parameters. The results confirm that increasing model complexity decreases the statistical significance of TDA estimates. p-values for LOO with MNIST3: 0.331 (2-layer CNN) → 0.370 (3-layer CNN) → 0.786 (ViT+LoRA). Exp. 4 in global response.
> ### How can [successful work with IFs] be reconciled with the results?
> We share the reviewer’s intuition that successful related work (e.g. [3,15]) could be explained via low-noise pairs. Due to the prohibitive cost of LOO, previous work relied on a small number of train-test pairs (100 [3], 500 [15]). We internally attempted to replicate successful TDA results [3,15] but found that the impact of training stochasticity generally dominates the impact of a single training sample. This observation motivated our work.
> ### Eq. 10[...] I assume these hypotheses refer to τ’s mean [...]. Is this correct?
> The reviewer is correct. The hypothesis should read μ=0 where μ is the mean of the distribution τ(zj,z). We will correct this, thanks!
> ### Eq. 11 what version of the T-test are you using?[...] Is the test designed to consider the pairing of the posterior (and perturbed posterior) samples?
> We use the (unpaired) Student’s t-test [a] to test the statistical significance of μ>0, where μ is the mean of TDA estimates. In Eq. (9), we do not pair the indices $t$ and $t^\prime$ for the two posteriors.
> ### Is it common to combine these posterior sampling techniques?
> Yes, e.g. [b] uses both DE and SWA to parametrise the posterior. This implies a mixture-of-Gaussian posterior family, where the DE models the centroids and SWA models each Gaussian component.
> ### The results would be more convincing if the experiments had focused on only a subset of the test-train pairs, but considered larger [...] dataset sizes.
> We ran an experiment with a CNN on MNIST to study TDA reliability on a set of 1000 train-test pairs. The results are in line with our submission: TDA is inherently stochastic and TDA methods fail to capture the variance in LOO. Exp.1 in global response.
> ### Other
> We will add [c], increase label sizes in Fig. 3 and 4, and include the dataset condition in Eq. 7.\
> [a] Student (1908)\
> [b] Wilson&Izmailov (NeurIPS 2020)\
> [c] Hampel (1974)

---

> > ### Comment · Reviewer_fqXw · 2023-08-15
> >
> > I appreciate the authors taking the time to write a detailed rebuttal and to provide additional experiments. However, many of my concerns remain unaddressed, e.g.,
> >
> > "it is unclear which additional theoretical treatment would add value": I struggle with this response. As a researcher and practitioner who regularly uses these methods, I am saying that the paper hasn’t provided me with much insight beyond what I have already observed empirically or read in related works. I therefore question the extent of its value to the community. I agree that in practice TDA methods show a great deal of stochasticity. At least when applied to non-convex models optimized via SGD. Perhaps, what I’m struggling most with here is the framing. If the paper was simply positioned as saying: “User be warned, most TDA values in neural networks are dominated by retraining noise”, then I would be less critical. (Albeit variations of this point have been made before). But the paper is positioned as a “Bayesian perspective” on TDA, and as such I was expecting the subject to be investigated more theoretically, and I was hoping it might present some insights into the observed stochastic behaviors of these methods, e.g., provide a theoretical explanation for the training data points which are consistently high influence for a test point (what you call low-noise pairs). Moreover, what if the reader were interested in a simpler model class, like linear regression or logistic regression? These are important use cases for TDA methods. But influence functions are not “inherently stochastic” in linear regression (nor in logistic regression–assuming sufficient optimization). The practitioner would need to consider the application of the TDA methods to the Bayesian version of these methods, which aim to explicitly model uncertainty in the parameters. This is left unaddressed, e.g., how would one apply IF to simple Bayesian models (ones where parameter uncertainty is explicitly represented)?
> >
> > “Even in the deterministic setup, the IFT and inverse Hessians are generally inapplicable due to the non-convexity and high dimensionality of the deep learning optimization problem. They are not novel challenges introduced by our probabilistic treatment.” I agree that some TDAs already suffer theoretical issues when applied to neural networks. However, this is not a reason to avoid explicitly naming them and discussing their implications. In practice, these shortcoming are often circumnavigated by either continuing to train from an optimized checkpoint, or by keeping all but the last layer parameters frozen (e.g., Koh and Liang), effectively leading to a logistic regression model over a fixed feature extractor. Neither of these approaches are examined or discussed.
> >
> > “We add a damping term λ to ensure the positive definiteness (PD) of Hessian”: This damping term is an example of something that may be interesting to examine theoretically, could it be connected to the prior in a Bayesian learning framework?
> >
> > “we ran an experiment with a 3-layer CNN that matches ViT+LoRA in #trainable parameters. The results confirm that increasing model complexity decreases the statistical significance of TDA estimates ” : There are still only 3 observations here. When the dataset size experiments were increased from 3 observations to 6 observations the trend changed. What makes the authors believe that this claim about model complexity is more robust?
> >
> > I still struggle with the setup of the main statistical test. As far as I can tell, Equation 11 does not correspond to an unpaired, 1 sample student’s t-test. Assuming the p-values were nonetheless correctly calculated, it’s still unclear whether this test is appropriate given the mixed nature of the posterior samples. The samples are not IID. The “seed” samples from DE are arguably IID, but the subsequent samples from SWA are not. This should at least be discussed.
> >
> > As such I stand by my original score.

---

> > > ### Author Response · Authors · 2023-08-18
> > > **Response to reviewer (1/2)**
> > >
> > > We thank the reviewer for the follow-up comments and suggestions that will improve the quality of our paper.
> > >
> > > > I am saying that the paper hasn’t provided me with much insight beyond what I have already observed empirically or read in related works [...] Perhaps, what I’m struggling most with here is the framing. [T]he paper is positioned as a “Bayesian perspective” on TDA, and as such I was expecting the subject to be investigated more theoretically.
> > >
> > > Thank you for the clarification and we are glad to confirm that the reviewer shares a similar experience regarding the stochasticity of TDA. We fully agree that our framing is broader than our main focus and this is potentially a critical issue. We propose to address this issue by changing the title to delineate the scope more precisely and convey the message more concretely: “A Bayesian Approach to Analysing Training Data Attribution in Deep Learning”. We will also update the abstract and introduction to make clear that our scope is the application of TDA on deep learning models.
> > >
> > > We would like to clarify further that we do not aim for a theoretical contribution. We study how the stochasticity in the training process of deep models fluctuates the TDA estimates. We apply prominent Bayesian deep learning approaches like Deep Ensemble and Stochastic Weight Averaging to model the randomness in the deep learning training procedure. We find that training process stochasticity has a considerable effect on TDA, as we observe significant variances in the scores. We further find that the tested approximation methods fail to fully capture the ground-truth variance. Based on this observation, we recommend evaluating the approximation methods for not only replicating the mean leave-one-out (LOO) measure but also the variance therein. We do such an evaluation for not only influence function (IF) but also other approximate TDA techniques like additional training step (ATS), grad-dot (GD) or TracIn, and grad-cos (GC).
> > >
> > > > I was hoping it might present some insights into [...] e.g., provide a theoretical explanation for the training data points which are consistently high influence for a test point [and] how would one apply IF to simple Bayesian models [...]. Moreover, what if the reader were interested in a simpler model class, like linear regression or logistic regression?
> > >
> > > We agree that probing the link between sample traits and TDA noise is important. However, even for simple model classes, characterizing TDA value distribution is inherently complex.
> > >
> > > For example, even if we assume a simple Gaussian posterior for the learned parameter $p(\theta| X)=\mathcal{N}(\mu,\Sigma)$, the resulting IF value for logistic regression corresponds to
> > >
> > > $$ IF(z_\text{test},z) -y_\text{test} y \cdot \sigma(-y_\text{test}\theta^\top x_\text{test})\cdot\sigma(-y\theta^\top x)\cdot x_\text{test}^\top H_\sigma^{-1} x$$ (Koh & Liang 2017).
> > >
> > > where $\sigma(\cdot)$ is the sigmoid function. Note that the product of the sigmoids of Gaussian random variables does not yield a tractable distribution that can be represented with a closed-form mean and variance formulae.
> > >
> > > Likewise, linear regression with an $\ell^2$-loss results in the following formula for IF:
> > >
> > > $$ IF(z_\text{test},z) = - (x_\text{test}^T  H^{-1} x) (x_\text{test} \cdot \theta - y) (x\cdot \theta - y) $$
> > >
> > > Here, again, the product of two Gaussian random variables does not yield a distribution with tractable mean and variance, making it rather non-trivial to analyse the theoretical behaviour.
> > >
> > > > influence functions are not “inherently stochastic” in linear regression (nor in logistic regression–assuming sufficient optimization)
> > > We agree that we need to be careful with this statement and that influence functions are not “inherently stochastic”. We apologise. In the paper, we will explicitly restrict our scope to deep learning models.
> > >
> > > > However, this is not a reason to avoid explicitly naming them and discussing their implications.
> > >
> > > Yes, agreed. We will discuss the mentioned issues with influence functions in the manuscript in §3, under paragraph *TDA methods likewise estimate random quantities*.

---

> > > ### Author Response · Authors · 2023-08-18
> > > **Response to reviewer (2/2)**
> > >
> > > > In practice, these shortcoming are often circumnavigated by either continuing to train from an optimized checkpoint, or by keeping all but the last layer parameters frozen (e.g., Koh and Liang), effectively leading to a logistic regression model over a fixed feature extractor. Neither of these approaches are examined or discussed.
> > >
> > > The mentioned approaches (i.e. continuing to train from optimized parameter, freezing the model up to the last layer) reduce the stochasticity in the parameter. However, as the reviewer would be aware from _Bae et al._’s work (_If Influence Functions are the Answer, Then What is the Question?_), these approaches lead to a gap between influence scores and what they set out to measure, which is the leave-one-out retraining (LOO). In our study, we aim to explore the variance in TDA scores based on the original LOO definition, and observe how various TDA approximation methods capture this variability.
> > >
> > > > “We add a damping term λ to ensure the positive definiteness (PD) of Hessian”: This damping term is an example of something that may be interesting to examine theoretically, could it be connected to the prior in a Bayesian learning framework?:
> > >
> > > The damping term could indeed be seen as an isotropic Gaussian prior centred at the origin. We thank the reviewer for pointing this out and will add this comment to the paper.
> > >
> > > > When the dataset size experiments were increased from 3 observations to 6 observations the trend changed.
> > >
> > > The relation between training set size and the variability of TDA scores is indeed not linear, and we will update §4.3 *Training set size* for the final version of the paper. We would like to correct this observation by stating that an increased training set size leads to an increased variability in TDA scores *up until a certain point*. We observe that TDA scores tend to be smaller with larger training sets:
> > >
> > > | Training set size | Mean TDA score (LOO) | Mean variance of LOO |
> > > | --- | -------------------- | -------------------- |
> > > | 30  | 0.242                | 0.098                |
> > > | 60  | 0.042                | 0.041                |
> > > | 90  | 0.048                | 0.093                |
> > > | 120 | 0.073                | 0.091                |
> > > | 150 | 0.030                | 0.019                |
> > > | 180 | 0.017                | 0.019                |
> > >
> > > We believe this makes sense because a single training sample tends to attribute less to test samples overall, as the training set size increases. If all train-test pairs consistently exhibit low TDA scores, the variance in scores decreases, resulting in smaller p-values, which we observe as the drop in p-values for larger training sets, such as $|D|\in{150,180}$  in additional experiment 3 of the rebuttal pdf.
> > >
> > > > What makes the authors believe that this claim about model complexity is more robust?
> > >
> > > We believe that our claim about model complexity (more complex models tend to exhibit larger variability in their TDA scores) is robust to changes in the observed trend because we consistently observe that model complexity mainly affects the variance in TDA scores more strongly than the mean score itself:
> > >
> > > |      Model       | Mean TDA score (LOO) | Mean LOO variance |
> > > | ----------- | -------------------- | ----------------- |
> > > | 2-Layer CNN | 0.030                | 0.019             |
> > > | 3-Layer CNN | 0.047                | 0.045             |
> > > | ViT+LoRA    | 0.058                | 0.224             |
> > >
> > > This observation aligns with the intuition of using a mostly frozen model to compute IFs (Koh & Liang, 2017): With more parameters, we increase the stochasticity of the training procedure, in turn diminishing the reliability of TDA scores.
> > >
> > > >  As far as I can tell, Equation 11 does not correspond to an unpaired, 1 sample student’s t-test.
> > >
> > > We use a 1-sample Student’s t-test for a single random variable, TDA score $\tau$. There is no pairing of samples. Note that $z_j$ and $z$ are fixed variables in Equation 11.
> > >
> > > > The samples are not IID. The “seed” samples from DE are arguably IID, but the subsequent samples from SWA are not. This should at least be discussed.
> > >
> > > Sure, for the mixture-of-Gaussian posterior, we do not sample IID. Instead, we perform a version of stratified sampling, where we fix the number of samples from each centroid. Within each centroid, the sampling is IID, as the reviewer has pointed out. We believe that the stratified sampling approach used in this study is unlikely to introduce significant bias in our statistical analysis. This is because we have ensured an equal number of samples from each stratum, and the strata (centroids) themselves are exchangeable, meaning that their order does not affect the overall outcome of the analysis. We thank the reviewer for this critical observation. We will include this discussion.

---

### Official Review · Reviewer_fmRD · 2023-07-06

**Soundness:** 4 excellent
**Presentation:** 4 excellent
**Contribution:** 3 good
**Rating:** 7
**Confidence:** 3

**Summary:**

This paper presents a Bayesian perspective on Training Data Attribution (TDA), a technique that identifies influential training data for model predictions. The authors propose treating the learned model as a Bayesian posterior and TDA estimates as random variables. This approach reveals that the influence of individual training data often gets overshadowed by noise from model initialization and batch composition. Consequently, they suggest using TDA only when certain training data consistently influences model predictions, despite noise factors.

**Strengths:**

- Well written, easy to follow. Thanks!
- Studies an important practical problem and provides a practically relevant recommendations.
- Strong analysis of experiments

**Weaknesses:**

- The paper identifies difficulties with current practice and provides some recommendations, it does not discuss a solution or paths towards a solution of the TDA problem.

**Questions:**

l 65: what do you mean by "more global"?

**Limitations:**

n.a.

---

> ### Author Rebuttal · Authors · 2023-08-10
>
> We appreciate the encouraging review.
>
> In the following, we answer the questions posed by the reviewer one by one:
>
> ### [The paper] does not discuss a solution or path towards a solution of the TDA problem.
> Our focus is to identify issues with prior problem definition of the TDA task and to propose a novel problem definition that makes more sense in practice. In the process, we have shared interesting intuitions that may lead to effective solutions. For example, we have identified varying degrees of variance in different train-test pairs (Figure 3) and have recommended that future researchers study the low-noise train-test pairs, where the TDA estimates are stable (Section 4.5, line 298). Another path towards a more practical TDA technique is to study the factors affecting the inherent variance of TDA.
> We will make this discussion more explicit in the final version of the paper.
>
> ### Line 65: What do you mean by “more global”?
> We categorise a TDA method as local vs global based on the expected counterfactual impact of “altering a training sample $z_j$”. For example, the leave-one-out (LOO) re-training introduces a global impact on the model parameters, as the model is trained for multiple iterations without the training sample $z_j$. On the other hand, taking a single additional training step on $z_j$ is considered local, as the expected impact is more restricted.
>
> We understand that this is not a well-defined terminology. We will edit the sentence as follows:
> “(5) Observation that the TDA estimation methods capture local changes in the model with regard to the counterfactual question of “retraining without $z_j$”, while LOO retraining itself results in a more global change through the training procedure.”
> We are happy to discuss this point and hope to find a phrasing that will improve the paper for better clarity here.

---

> > ### Comment · Reviewer_fmRD · 2023-08-14
> > **Thank you**
> >
> > I appreciate the explanation.

---

### Official Review · Reviewer_CbTs · 2023-07-06

**Soundness:** 3 good
**Presentation:** 4 excellent
**Contribution:** 3 good
**Rating:** 6
**Confidence:** 4

**Summary:**

In this paper the authors investigate training data attribution (TDA) through a bayesian lens by explicitly considering the randomness in estimating the model parameters with and without a given training example.  To generate approximate bayesian posteriors on model parameters the authors use deep ensembles and SWA, then consider multiple different methods of TDA. They find that, using a t-test, TDA estimates often exhibit high variance, indicating high signal to noise ratio. This variability is often dominated by noise from model initialization and seems to increase with training set size and model complexity. They also show strong consistency between different groups of TDA methods.

**Strengths:**

The manuscript is well written and clear, and the problem setting important. Measuring the variability in downstream TDA estimates caused by variability in model parameters is an interesting perspective that adds to the growing body of literature aiming to understand various methods of TDA. The conclusions drawn by the authors regarding sources of randomness (initialization vs batching) seem to align with current intuitions around TDA estimates and present a new way to measure the quality of any proposed TDA method by considering those pairs for which the LOO estimates are statistically significant.

**Weaknesses:**

One of the weaknesses is the small dataset sizes considered. The authors restrict datasets to 150 training examples and 900 test examples for MNIST and 500 train/test examples for CIFAR10. At these small dataset sizes, we might expect higher variability and correlation between training and test examples. For example, if the model only sees a single 3 written a particular way during training, then we might expect high variability in differently trained model’s predictions at a similar 3 in the test set. However, seeing many such 3s in a larger dataset should decrease the variability of posterior predictions. Although the conclusions drawn and trends observed at these small scales are interesting, it remains to be seen whether they hold for larger datasets. As a starting point, the authors should consider running the same experiments on a larger training set but consider only a small random subsample in their analysis. This should give a stronger signal for the behavior of the random variable $\tau$ for larger datasets.

The other main set of experiments that would greatly strengthen the paper is the consideration of the downstream tasks for which TDA is useful, for example in mislabel identification. Since these tasks may not depend on exact LOO estimates [1], the inherent variability in TDA estimates may or may not be a relevant consideration. It would be interesting to see the extent to which the variance of $\tau$ estimates makes these tasks difficult or even potentially ill-posed.

[1] Juhan Bae, Nathan Ng, Alston Lo, Marzyeh Ghassemi, and Roger B Grosse. If influence functions are the answer, then what is the question?

**Questions:**

- Is there a hypothesis for why so many more MNIST3 examples seem to have low p-values compared to CIFAR10 examples? It would be interesting to see whether these low p-value examples have some special properties that could be exploited to identify them efficiently.
- It seems that the model complexity analysis should be considered relative to the dataset we are attempting to perform TDA on. Is there any way to quantify this value? Although a ViT model may exhibit high $\tau$ variability on a simple dataset like CIFAR10 or MNIST3, it may behave differently on a much more complex  datasets like ImageNet.

**Limitations:**

Yes

---

> ### Author Rebuttal · Authors · 2023-08-10
>
> We thank the reviewer for the thorough review and recommendation to accept our work.
>
> We wish to address the remarks and questions raised by the reviewer one by one:
>
> ### One of the weaknesses is the small dataset sizes considered.
>
> We understand that considering greater dataset sizes would be desirable. However, the experiments are already computationally heavy when it comes to TDA evaluations that involve re-training of the whole network several times, each time leaving out a single training sample.  We have retrained the model #train $\times$ #MC samples $=150\times 50=7500$ times for the MNIST3 experiments and $500\times 50=25000$ times for the CIFAR10 experiments. This surmounts to roughly 2 and 104 Nvidia 2080ti-hours per single TDA evaluation. This explains why TDA community has relied on apparently smaller-scale experiments: [a,b] have considered only 100 and 500 train-test pairs, respectively. In this work, we opt to work with smaller datasets and thorough consideration of all train-test pairs $(z_j,z)$. We believe that the key problem with previous TDA papers is the selection of certain train-test pairs to report the results on. Thanks to the exhaustiveness, we are capable of making conclusions based on the entire population of train-test pairs: we verified the prevalence of high-noise train-test pairs and the existence of very few low-noise pairs (Section 4.5, line 294).
>
> ### It remains to be seen whether they hold for larger datasets. As a starting point, the authors should consider running the same experiments on a larger training set but consider only a small random subsample in their analysis.
> We ran an additional experiment following the reviewer’s suggestion: We trained a model on the full MNIST dataset and defined a smaller subset of train-test pairs (100$\times$10=1000) for studying TDA. Note that MNIST is not small for our experimental setting, where we compute LOO retraining. In fact, training the model once on one Nvidia 2080ti GPU cost around 7 min, which we did 100$\times$50 posterior samples = 5000 times, resulting in roughly 583 GPU hours. We confirm the main findings in the smaller-scale experiments: the ground-truth TDA values tend to show high (p>0.05) variance in general. Experimental details in the global response, experiment 1.
>
> ### The other main set of experiments that would greatly strengthen the paper is the consideration of the downstream tasks for which TDA is useful, for example in mislabel identification.
>
> Following the reviewer’s suggestion, we performed a mislabel identification experiment similar to [a] and [c]. We find that the inherent stochasticity of TDA leads to a large range in mislabel identification performance and that high variance in the TDA estimates degrades downstream task performance. Experimental details in the global response, experiment 2.
>
> ### Is there a hypothesis for why so many more MNIST3 examples seem to have low p-values compared to CIFAR10 examples?
>
> Our intuition is: MNIST3 is an easier task to learn than CIFAR10, which could result in a less complex loss landscape where there may be global optima that are easier to find. If model posteriors are sampled from the same optimum there could be less variation in the TDA scores.
>
> ### It would be interesting to see whether these low p-value examples have some special properties that could be exploited to identify them efficiently.
>
> We agree that a study on the properties of low-noise train-test pairs would be interesting, which is also one of our main recommendations to the community. We do not have a strong hypothesis for the cause of this phenomenon yet, as the noise level seems independent of obvious features like classifier confidence. This is an interesting research question to be explored in the future.
>
> ### It seems that the model complexity analysis should be considered relative to the dataset we are attempting to perform TDA on. Is there any way to quantify this value? Although a ViT model may exhibit high 𝜏 variability on a simple dataset like CIFAR10 or MNIST3, it may behave differently on a much more complex datasets like ImageNet.
>
> We agree that it makes sense to match the model complexity to the data complexity. We applied LoRA finetuning to reduce the effective model complexity of a ViT down to match the CIFAR10 dataset. Though it would be interesting to perform our analysis on ViTs trained on ImageNet, this is computationally prohibitive - it would require $10,000\times 50 = 500,000$ times retraining the ViT, which roughly corresponds to 2.4 GPU years. A realistic research scale at the moment is what we have presented.
>
> [a] Understanding black-box predictions via influence functions (ICML 2017)
>
> [b] FastIF (EMNLP 2021)
>
> [c] Estimating training data influence by tracing gradient descent (NeurIPS 2020)

---

> > ### Comment · Reviewer_CbTs · 2023-08-18
> >
> > Thanks for the detailed response. After reading the other reviewer responses and thinking more about this work myself I have a few more reservations.
> >
> > The additional experiments address some of my concerns but it is still difficult to support the claims made since the noise could be stemming from many different sources that are difficult to disentangle in these settings. Although, I recognize the difficulty in attaining gold-standard LOO retraining TDA estimates, we know from [2] that LOO does not necessarily correlate with IF (and the other methods as well given the analysis in Figure 5), so comparing the two is a bit misleading, especially as neural network sizes grow larger and this discrepancy increases. One option is to use PBRF training, as in [1, 2], which we know corresponds well with IF. Applying the same analysis to this training regime would give us a more clear picture of the stochasticity of IF. As it stands, the observed variability is not so surprising when we consider what the existing TDA methods measure as compared to true LOO retraining.
> >
> > Another consideration is that TDA methods are often used to analyze the predictions of a **specific** model, rather than a general class of models.  In these cases we care only about the specific sample from the unknown model posterior and not the posterior itself. The suggested analysis above of the PBRF training regime would more closely align with this problem setting. In light of these concerns I will be adjusting my score down one point.
> >
> > [1] Studying Large Language Model Generalization with Influence Functions. Roger Grosse ̊:, Juhan Bae ̊:, Cem Anil ̊ et al. 2023.
> > [2] Juhan Bae, Nathan Ng, Alston Lo, Marzyeh Ghassemi, and Roger B Grosse. If influence functions are the answer, then what is the question?

---

> > > ### Author Response · Authors · 2023-08-20
> > >
> > > We thank the reviewer for the great additional points to the original review.
> > >
> > > > [...] it is still difficult to support the claims made since the noise could be stemming from many different sources that are difficult to disentangle in these settings.
> > >
> > > We agree that in practice, the source of noise will be diverse and potentially difficult to disentangle. But in our experimental setting, we used random initialisation and batch composition as sources of noise, which could effectively be controlled:
> > > “In a variant of DE with the initialisation as the source of randomness (\textbf{DE-Init}), we train each of $T_\text{DE}$ randomly initialised parameters $\theta^{(t)}\_0$ on either $\mathcal{D}$ or $\mathcal{D}\_{\setminus j}$. (...) We also consider the batch composition in stochastic gradient descent (SGD) as the source of randomness (\textbf{DE-Batch}). (...)” [§4.1, lines 181-186]
> > > Controlling only these two factors are sufficient to support the claim that training process stochasticity (via random initialisation and batch composition) leads to variance in TDA scores [§4.5, lines 287-288].
> > >
> > > > [...] we know from [2] that LOO does not necessarily correlate with IF [...], so comparing the two is a bit misleading, especially as neural network sizes grow larger and this discrepancy increases.
> > >
> > > We thank the reviewer for initiating an interesting discussion around the seminal work of Bae et al. (2022). We are aware that IF (one of the tested TDA methods in this work) does not exactly correspond to LOO from Bae et al. (2022) and the discrepancy increases with larger networks, as the reviewer has described. Nevertheless, we strongly believe that the ultimate goal of TDA is to predict the counterfactual outcome of removing a training sample, rather than the modified PBRF objective in Bae et al. (2022). We thus argue that it is necessary to make an empirical comparison between the ground-truth LOO and approximate TDA methods like IF, precisely to quantify the aforementioned discrepancy. [§4.5, lines 291-293].
> > > Please let us know if there is any misunderstanding on the reviewer’s comments from our side.
> > >
> > > > [...] the observed variability is not so surprising [considering] what the existing TDA methods measure as compared to true LOO retraining.
> > >
> > > We are not completely sure if we understand the reviewer’s comment correctly. Our understanding of the reviewer’s comment is: because existing TDA methods are very crude approximations of the true LOO retraining, it is indeed unsurprising that TDA methods show great variability. We believe there could be a bit of confusion. Our main point is that even the true LOO retraining, which is considered the ground-truth target for approximate TDA methods, exhibits stochasticity: “Generally, we observe many TDA measurements, ground-truth and estimations likewise, are unstable with non-significant p-values (> 0.05).” [§4.2, lines 204-205].
> > > Because the ground-truth target is stochastic, our point is that we need to adjust our evaluation protocol to embrace the inherent stochasticity of the task and treat TDA values as random variables [§4.5, lines 287-293]. We measure the Pearson and Spearman correlations of TDA estimate’s mean and variance to study how well approximate TDA methods capture the ground-truth LOO scores in both mean and variance. We’d be happy to discuss further if we have not understood the reviewer’s comment correctly.
> > >
> > > > Another consideration is that TDA methods are often used to analyze the predictions of a specific model, rather than a general class of models. In these cases we care only about the specific sample from the unknown model posterior and not the posterior itself.
> > >
> > > This is a great point. Indeed, in practice, the starting point would be a fixed trained model. It would also make sense to use a fixed model trained on the original dataset $\theta_\mathcal{D}$, rather than the posterior $\theta\sim p(\theta|\mathcal{D})$. However, to answer the question “how does the model output change when a sample $z_j$ is removed from the training set?”, we should inevitably introduce multiple possibilities for the counterfactual model $\theta_{\mathcal{D}\setminus j}$. There is no well-defined notion of a *unique* trained model obtained from re-training a model without a certain sample because the exclusion distorts the batch composition, for example. Such an ambiguity in $\theta_{\mathcal{D}\setminus j}$  is well-captured via Bayesian model posterior $\theta\sim p(\theta|\mathcal{D}_{\setminus j})$. We extend this posterior viewpoint to the model built from the original dataset $\theta\sim p(\theta|\mathcal{D})$, but one could still consider treating this as a fixed variable. We will discuss this option in the final version.
> > >
> > > We are glad to be able to continue the constructive discussion. We will incorporate the discussion here to the final manuscript. In case there is any misunderstanding from our side or the reviewer has any follow-up questions, we will remain available.

---

### Author Rebuttal · Authors · 2023-08-10

We thank the reviewers for their constructive comments and suggestions. Reviewers agree that we “study an important practical problem” (fmRD, CbTs) and provide “thorough experiments” (wqEs) and a “strong analysis” (fmRD).

We have addressed individual reviewers’ comments and questions in the dedicated responses. We use the global response to show the results of experiments requested by the reviewers.

### 1: Larger dataset (fqXw, CbTs)

Summary: We perform the test for statistical significance of TDA given a model trained on MNIST ($|D|=60,000|$). We base the statistical analysis on a small random subsample and find high (p>0.05) variance in ground-truth TDA as well as TDA estimates, in line with previous findings.

Setting:
We train a 2-layer CNN on the full MNIST dataset and sample $T=50$ times from the posterior. For analysis of TDA methods, we uniformly sample 100 $z_j$ from the training set and 10 $z$ from the test set, resulting in 1000 train-test pairs.
Results:
The model has a predictive accuracy on the MNIST test set of $0.979 \pm 0.001$ at 95% CI. The p-values resulting from the TDA analysis are:
LOO  |ATS  |IF   |GD   |GC
-----|-----|-----|-----|-----
0.761|0.362|0.464|0.475|0.247

Distribution of p-values for LOO and correlation analysis is attached in the global response PDF.

The results show that TDA estimates vary strongly for the small subset of train-test pairs (high p-values). Given our previous findings, we find two main reasons: (1) MNIST is larger in training set size than the initial sets we used. The attribution of one sample to model behavioris likely to be marginal and unstable. (2) Low-noise samples exist, but they are in the minority. This experiment shows that LOO is inherently noisy and TDA approximation methods fail to capture this. This conclusion aligns with the results in the submission.

### 2: Mislabel identification (wqEs, CbTs)

Summary: We intentionally mislabel parts of the training dataset and aim to identify the mislabeled samples using TDA. We find that the inherent stochasticity of TDA leads to a large range in mislabel identification performances.

Setting:
We perform this experiment with the 2-layer CNN and MNIST3 ($|\mathcal{D}|=150$) and CIFAR10 ($|\mathcal{D}|=500$). We follow the procedure from Koh & Liang [a]: First, a random 10% of the datasets are mislabeled by the highest scoring incorrect label. We train the model using these mislabeled datasets (sample $T=50$ times from the posterior). Then, we compute self-influence, which is the attribution of a sample to itself $\tau(z_j, z_j)$ [a], with each TDA method. The mislabeled dataset is ranked according to self-influence and the quantity of interest is the fraction of mislabeled samples found when inspecting the top x% of the ranked dataset.
In the analysis, we inspect (1) the range of mislabeled fractions discovered if we treat TDA deterministically, i.e. we compute the discovered fraction per $t\in T$ and report the range; (2) the fraction of mislabeled samples discovered when we use the mean over the TDA scores of our posterior samples.

Results:
The results plots are visualized in the global response PDF.

We find that deterministic TDA results in a large range of possible outcomes for identifying mislabeled samples. This means that it is harder to reliably identify mislabels when TDA is treated as point estimates.

### 3: Training set size (fqXw)

Summary: Ablations with training set sizes, expanding on the experiment from section 4.3 “Training set size”, show that there may be a point of “stochasticity saturation” where the increasing size of the trainset does not contribute more to the variance in TDA estimates.

Setting:
We expand the experiment with the 2-layer CNN on MNIST3 and CIFAR10 from section 4.3 Training set size by testing the model trained on MNIST3 of sizes 90, 120, 180 and CIFAR10 of sizes 300, 400, 600.

Results:
Updated plots of Figure 4 in global response PDF.

The results indicate that an increase in training set size does not necessarily lead to an increase in noise (i.e. the relationship is not linear). Instead, we observe that there may be a point of “stochasticity saturation” where the increasing size of the training set does not contribute more to the variance in TDA estimates (p-values stay large, but don’t increase).
We will update the discussion in paragraph 4.3 “Training set sizes” to reflect this for the final version.

### 4: Model complexity (fqXw)
Summary: An additional experiment with a 3-layer CNN shows that model complexity is likely a factor in the reliability of TDA estimates.

Setting:
We train a 3-Layer CNN with 620,362 trainable parameters on MNIST3 ($|\mathcal{D}|=150$), CIFAR10 ($|\mathcal{D}|=500$) to analyse TDA methods. We chose this model as it is comparable to the 2-layer CNN in terms of architecture and to the ViT+LoRA model in terms of trainable parameters (597,514).

Results:
P-values (Distributions of p-values for ground-truth TDA (LOO) in the global response PDF.)
p-values|LOO  |ATS  |IF   |GD   |GC
--------|-----|-----|-----|-----|-----
MNIST3  |0.370|0.368|0.464|0.470|0.005
CIFAR10 |0.687|0.432|0.579|0.581|0.365

The results show that model complexity is likely a factor for the stochasticity in TDA estimates, where increasing model complexity (architecture and number of trainable params) means increasing variance, in line with previous findings. We also note that the number of low-noise train-test pairs decreases with increasing model complexity, but they still exist.

[a] Understanding black-box predictions via influence functions (ICML 2017)

---

### Decision · Program_Chairs · 2023-09-21

**Decision:**

Accept (poster)

**Comment:**

This paper has a very simple message that reviewers found unsurprising/intuitive, but nevertheless appreciated. Reviewers took issue with the "Bayesian" framing, and I tend to agree, as it is more of a probabilistic perspective than anything else. I am accepting the paper, but the authors should make sure to (a) appropriately reframe the paper based on reviewer comments, and (b) carefully address all of the detailed comments, especially the ones raised by fqXw.